Corrected: Publisher correction

# Internalized TSH receptors en route to the TGN induce local G$_s$-protein signaling and gene transcription

Amod Godbole[1,2], Sandra Lyga[1,2], Martin J. Lohse[1,2,5] & Davide Calebiro [ID] [1,2,3,4]

A new paradigm of G-protein-coupled receptor (GPCR) signaling at intracellular sites has recently emerged, but the underlying mechanisms and functional consequences are insufficiently understood. Here, we show that upon internalization in thyroid cells, endogenous TSH receptors traffic retrogradely to the trans-Golgi network (TGN) and activate endogenous G$_s$-proteins in the retromer-coated compartment that brings them to the TGN. Receptor internalization is associated with a late cAMP/protein kinase A (PKA) response at the Golgi/ TGN. Blocking receptor internalization, inhibiting PKA II/interfering with its Golgi/TGN localization, silencing retromer or disrupting Golgi/TGN organization all impair efficient TSH-dependent cAMP response element binding protein (CREB) phosphorylation. These results suggest that retrograde trafficking to the TGN induces local G$_s$-protein activation and cAMP/ PKA signaling at a critical position near the nucleus, which appears required for efficient CREB phosphorylation and gene transcription. This provides a new mechanism to explain the functional consequences of GPCR signaling at intracellular sites and reveals a critical role for the TGN in GPCR signaling.

[1] Institute of Pharmacology and Toxicology, University of Würzburg, Würzburg, 97078 Germany. [2] Bio-Imaging Center/Rudolf Virchow Center, University of Würzburg, Würzburg Germany. [3] Institute of Metabolism and Systems Research, University of Birmingham, Birmingham, B15 2TT UK. [4] Centre of Membrane Proteins and Receptors (COMPARE), Universities of Birmingham and Nottingham, Nottingham, NG7 2RD UK. [5] Present address: Max Delbrück Center for Molecular Medicine, Berlin, 13125 Germany. Amod Godbole and Sandra Lyga contributed equally to this work. Correspondence and requests for materials should be addressed to D.C. (email: davide.calebiro@toxi.uni-wuerzburg.de)

G-protein-coupled receptors (GPCRs) constitute the largest family of receptors for hormones and neurotransmitters[1,2]. They are implicated in several human diseases and are major pharmacological targets[3]. For all these reasons, their signaling mechanisms have been intensively investigated.

Signaling by GPCRs is initiated by binding of an agonist to a receptor[1,2,4]. The ensuing conformational change in the receptor is then relayed via activation of heterotrimeric G-proteins to effectors located at the cell membrane[1,2,4]. Among these, adenylyl cyclases play a major role in GPCR signaling by generating the soluble second messenger cyclic AMP (cAMP), which is implicated in the regulation of a plethora of cellular functions, such as gene transcription and cell proliferation[1,2,4]. These effects of cAMP are mediated by activation of cAMP effectors, which include guanine nucleotide exchange proteins activated by cAMP (Epac), cyclic nucleotide-gated ion channels and protein kinase A (PKA)[5]. In particular, the translocation of PKA catalytic subunit

to the nucleus in response to GPCR activation regulates gene transcription via phosphorylation of transcription factors of the cAMP response element binding protein (CREB)/activating transcription factor family[6]. Although cAMP is a small diffusible molecule, it has been hypothesized that cAMP/PKA signaling is compartmentalized such that cAMP and/or PKA would exert their effects only close to their site of production/activation[5,7–11]. While this hypothesis is consistent with a number of experimental observations, a direct proof that cAMP/PKA microcompartments are involved in GPCR signaling is still missing[5,9].

Although several GPCRs rapidly internalize upon agonist stimulation and traffic through various intracellular compartments, signaling by GPCRs has long been believed to occur only at the cell surface. The only exception was represented by the G-protein-independent and β-arrestin-dependent activation of the mitogen-activated protein kinase pathway, which was shown to happen also on endosomal membranes[12]. However, recent

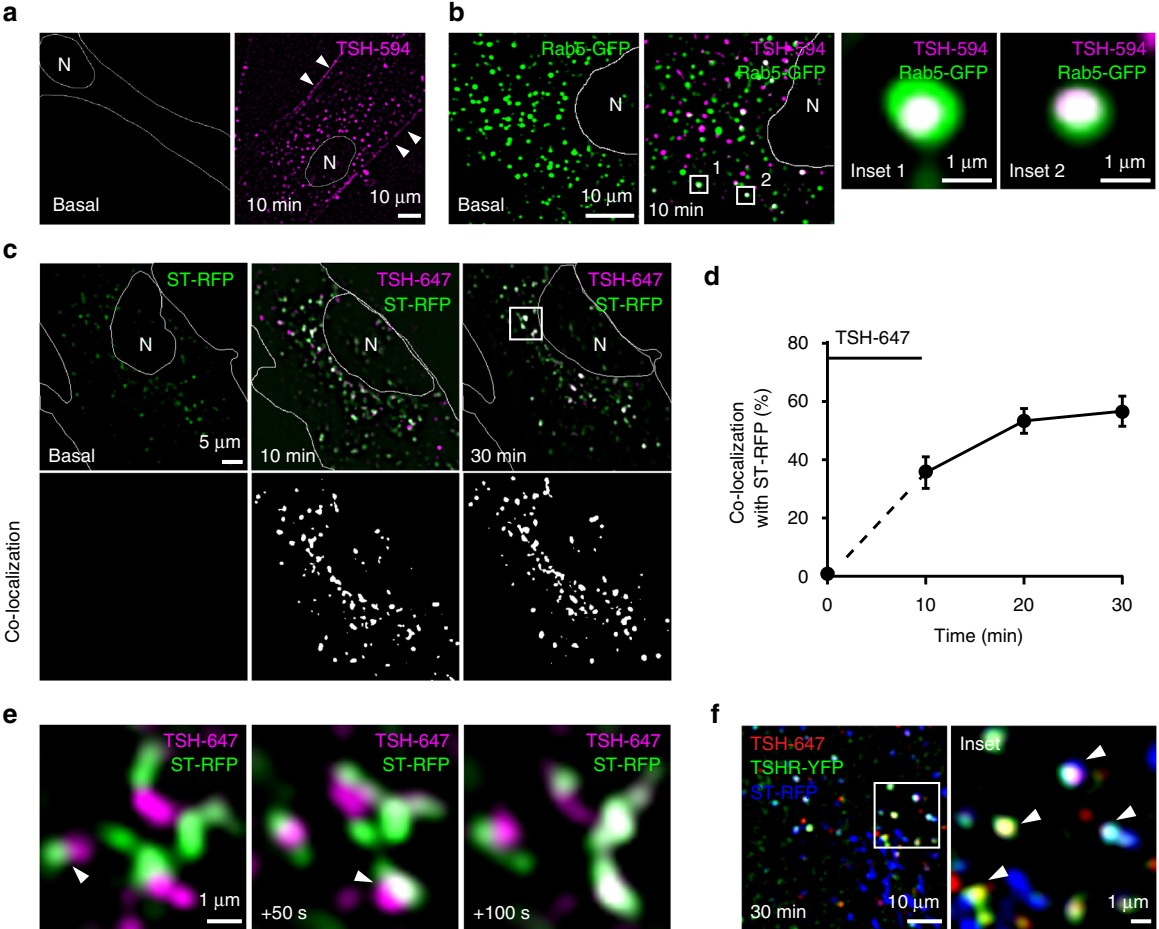

**Fig. 1** TSH/TSHR complexes traffic retrogradely to the TGN. Primary mouse thyroid cells were stimulated with fluorescent TSH (TSH-594 or TSH-647) for 10 min to visualize internalized TSH/TSHR complexes. When indicated, the cells were previously transfected with different combinations of subcellular markers. Shown are images at the indicated time points from beginning of stimulation. **a** Representative images of non-transfected cells obtained before stimulation and at 10 min, showing TSH bound to the plasma membrane and inside intracellular vesicles. The focus in the images shown in **a** was specifically set to also visualize TSH at the plasma membrane (arrowheads). **b** Simultaneous visualization of the early endosome marker Rab5-GFP and TSH-594. Representative images of the same cell obtained before and 10 min after stimulation are shown. Insets correspond to the regions marked with white boxes. **c** Time series showing accumulation of TSH-647 with the TGN-marker ST-RFP. White pixels in co-localization images indicate regions where TSH-647 and ST-RFP were simultaneously present. **d** Quantification showing the percentage (mean ± s.e.m.; n = 14 cells) of TSH-containing structures co-localizing with ST-RFP over time, calculated from images like those in **c**. **e** Enlarged view of the region delimited by the *white* box in **c**, showing individual fusion events (*arrowheads*). Time is given relative to the first image. **f** Simultaneous visualization of transfected TSHR-YFP, ST-RFP (*blue*) and TSH-647. Shown is a representative image acquired 30 min after stimulation. Inset corresponds to the region marked with the white box. White indicates triple co-localization. Arrowheads point to TSH/TSHR complexes in vesicles and tubules of the TGN. Where appropriate, cell edges and nucleus (N) are marked in white. Data in **a**–**c** and **f** are representative of 3, 3, 4 and 3 independent experiments, respectively

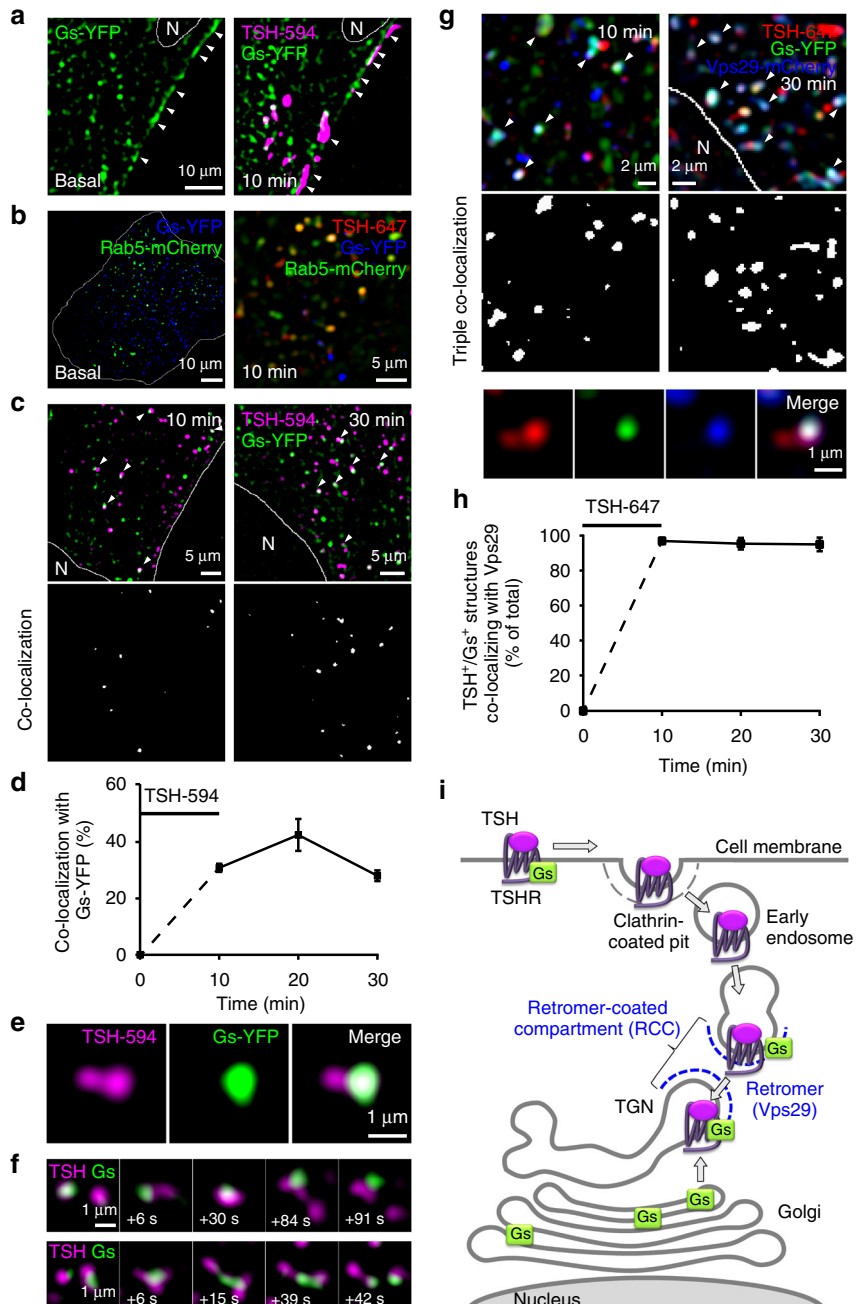

**Fig. 2** Internalized TSH/TSHRs meet the $G_s$-protein at the retromer-coated compartment (RCC) associated with the TGN. Primary mouse thyroid cells were cotransfected with different combinations of $G_s$-YFP and/or subcellular markers, followed by stimulation with fluorescent TSH (TSH-594 or TSH-647) for 10 min to visualize internalized TSH/TSHR complexes. Shown are images at the indicated time points from beginning of stimulation. **a** Co-localization between TSH-594 and transfected $G_s$-YFP at 10 min. The focus in the images shown in **a** was specifically set to also visualize TSH and the $G_s$-protein at the plasma membrane (arrowheads). **b** Co-localization between TSH-647, $G_s$-YFP and the early endosome marker Rab5-mCherry. Note the absence of triple co-localization after TSH stimulation. **c** Co-localization between TSH-594 and $G_s$-YFP. TSH is co-localizing with the $G_s$-protein on subdomains of tubulovesicular structures concentrated around the nucleus (arrowheads). **d** Quantification showing the percentage (mean ± s.e.m.; $n = 10$ cells) of TSH-containing structures co-localizing with $G_s$-YFP over time, calculated from images like those in **c**. **e** Enlarged view of a typical elongated membranous structure, showing presence of the $G_s$-protein on only one side and TSH on both. **f** Image sequences showing polarized fusion/splitting events among TSH- and $G_s$-protein-containing structures. Time is given relative to the first image. **g** Simultaneous visualization of TSH-647, $G_s$-YFP and Vps29-mCherry, used to localize the retromer complex. Structures analogous to those reported in **c** were observed (arrowheads). Bottom Enlarged view of a typical elongated membranous structure demonstrating the presence of retromer on the $G_s$-positive subdomain. **h** Quantification showing the percentage (mean ± s.e.m.; $n = 4$ cells) of structures containing both TSH-647 and $G_s$-YFP co-localizing with Vps29-mCherry over time, calculated from images like those in **g**. **i** Schematic view of the results. Where appropriate, cell edges and nucleus (N) are marked in white. Data in **a**–**c** and **g** are representative of 3, 3, 5 and 3 independent experiments, respectively

findings have challenged this classical model of GPCR signaling by suggesting that internalized GPCRs can activate G-protein-dependent pathways at intracellular sites[13–21].

The first direct evidence that internalized ligand/receptor complexes can stimulate cAMP production after their internalization came from studies of two typical protein hormone receptors: one by our group on the thyroid stimulating hormone receptor (TSHR)[13] and one by another group on the parathyroid hormone (PTH) receptor[14]. These studies independently suggested that GPCRs are able of inducing a second, persistent phase of cAMP production once internalized. Importantly, recent data suggest that signaling by internalized GPCRs is biologically relevant[21,22], as it has been implicated in the regulation of gene transcription[22] or even in the ultimate biological effects of a hormone[21]. However, the mechanisms linking GPCR signaling at intracellular sites to these biological effects are insufficiently understood[23–26].

Most of the available information regarding the site of intracellular GPCR signaling comes from experiments in which receptors have been overexpressed in heterologous cell lines. In these experiments, PTH[14] and β2-adrenergic[15] receptors have been found to co-localize with Gs-proteins on early endosomes. Moreover, a retrograde trafficking of PTH and vasopressin V2 receptors to the trans-Golgi network (TGN) has been shown to terminate cAMP signaling[20,27]. These data, together with direct evidence for the β2-adrenergic receptor using conformational biosensors[15], suggest early endosomes as the site of intracellular cAMP signaling. However, since trafficking mechanisms may differ substantially among cell types and might be affected by the level of expression of the receptor[28], it remains to be demonstrated whether the same mechanisms also operate on endogenous receptors in their native context. Furthermore, important differences might exist among individual receptors. Indeed, in our initial study on the TSHR, we found, based on immunofluorescence, co-localization between internalized TSH/TSHR complexes and Gs-protein in the Golgi/TGN[13]. This raises the intriguing possibility that, besides on early endosomes, GPCR signaling might also occur at other intracellular compartments.

In this study, we address these important questions by following the internalization/trafficking of an endogenous receptor in real time, using the TSHR in primary thyroid cells as a model, while directly visualizing the intracellular sites of Gs-protein activation and monitoring local cAMP/PKA signaling. We find that internalized TSHRs induce Gs-protein activation in a retromer-coated compartment (RCC) that is an extension of the TGN and mediates retrograde trafficking to this organelle, which is accompanied by a late phase of cAMP/PKA signaling at the Golgi/TGN. Interfering with receptor internalization, retrograde trafficking or the organization of the Golgi/TGN impair nuclear signaling in response to TSH. These data indicate a new potential role for the TGN as a central hub for GPCR trafficking and signaling and provide a new mechanism to explain the cellular effects of GPCR signaling at intracellular sites.

## Results

**Internalized TSHRs traffic retrogradely to the TGN.** To follow in real-time trafficking and signaling of endogenous TSHRs, we developed a highly sensitive imaging approach[29] based on fluorescently labeled TSH and highly inclined and laminated optical sheet (HILO) illumination[30]. This method is especially well suited to visualize trafficking and signaling events inside these cells, since they become flat and particularly thin (~1–2 μm) in culture[13]. Fluorescently labeled TSH, used to visualize TSH/TSHR complexes, conserved its biological activity[13] and bound specifically to TSHRs expressed in human embryonic kidney

(HEK293) cells (Supplementary Fig. 1) as well as to endogenous TSHRs in primary mouse thyroid cells (Fig. 1a and Supplementary Fig. 2). Immediately after stimulation of primary thyroid cells for 10 min, the fluorescent TSH was found on the plasma membrane (Fig. 1a and Supplementary Fig. 2, arrowheads) as well as inside intracellular vesicles/tubulovesicular strictures (Fig. 1a and Supplementary Fig. 2).

We then transfected primary mouse thyroid cells with different subcellular markers to characterize the compartment(s) involved in trafficking of the internalized TSH/TSHR complexes. Rab5[31] fused to GFP (Rab5-GFP) or a fragment of α-2,6-sialyl transferase fused to RFP (ST-RFP)[32] were used to identify early endosomes or the TGN, respectively. In cells stimulated with fluorescently labeled TSH, the internalized TSH was initially found in early endosomes (Fig. 1b), but it rapidly accumulated in perinuclear tubulovesicular structures labeled by the TGN marker ST-RFP, reaching ~60% of maximum already at 10 min (Fig. 1c, d and Supplementary Movie 1). After 30 min, ~60% of all internalized TSH was found in the TGN (Fig. 1c, d). This approach also allowed us to follow in real time the trafficking of TSH/TSHR complexes to the TGN, which occurred via fusion of TSH-containing vesicles with the tubulovesicular structures marked with ST-RFP (Fig. 1e and Supplementary Movie 2). Experiments performed in cells transfected with YFP-tagged TSHR (TSHR-YFP) confirmed that internalized TSH and TSHRs remained in the same compartment for at least 30 min after stimulation (Supplementary Fig. 3). Furthermore, to directly visualize the internalized TSH and TSHRs in the TGN, we cotransfected primary mouse thyroid cells with TSHR-YFP and ST-RFP, which were stimulated with fluorescently labeled TSH. The results confirmed the fast retrograde trafficking of TSH/TSHR complexes to the TGN (Fig. 1f). On the basis of the intensity and estimated volume of the TSH-containing vesicles, we calculated a TSH concentration in the lumen of the vesicles of ~ 800 nM (Supplementary Fig. 4). Since TSH has an affinity for its receptor in the picomolar range (high-affinity Kd = 169 pM)[33], we deduce that nearly all receptors in these vesicles should have TSH bound to them.

**Internalized TSHRs meet an intracellular pool of Gs-protein.** We then asked whether TSH/TSHR complexes and Gs-proteins cointernalize or meet subsequently in intracellular compartments. To answer this question, we performed a series of live-cell HILO imaging experiments in primary mouse thyroid cells transfected with a fluorescently tagged Gαs subunit (Gs-YFP) and stimulated with fluorescently labeled TSH. Under basal conditions, the Gαs subunit was found at the plasma membrane as well as on intracellular vesicles (Fig. 2a, left); only a minor fraction of these were early endosomes, visualized by cotransfection of Rab5-mCherry (Fig. 2b, *left*). Immediately after stimulation with fluorescent TSH, TSH/TSHR complexes were present at the plasma membrane (Fig. 2a, *right*) as well as on early endosomes (Fig. 2b, right); however, we found no Gs-protein on those early endosomes containing the internalized TSH/TSHR complexes (Fig. 2b, right). At the same time, TSH/TSHR complexes began to accumulate in a dynamic perinuclear tubulovesicular compartment in close association with the Gs-protein (Fig. 2c). The kinetics was similar to that of TSH accumulation in the TGN, reaching already ~70% of maximum after 10 min, when ~30% of the internalized TSH was found to co-localize with the Gs-protein (Fig. 2d). At a closer look, co-localization between TSH/TSHR complexes and the Gs-protein was observed on subdomains of this compartment. Figure 2e shows a typical example, where the Gs-protein is present on a protruding subdomain of an elongated membranous structure, whereas TSH/TSHR complexes are present, as often,

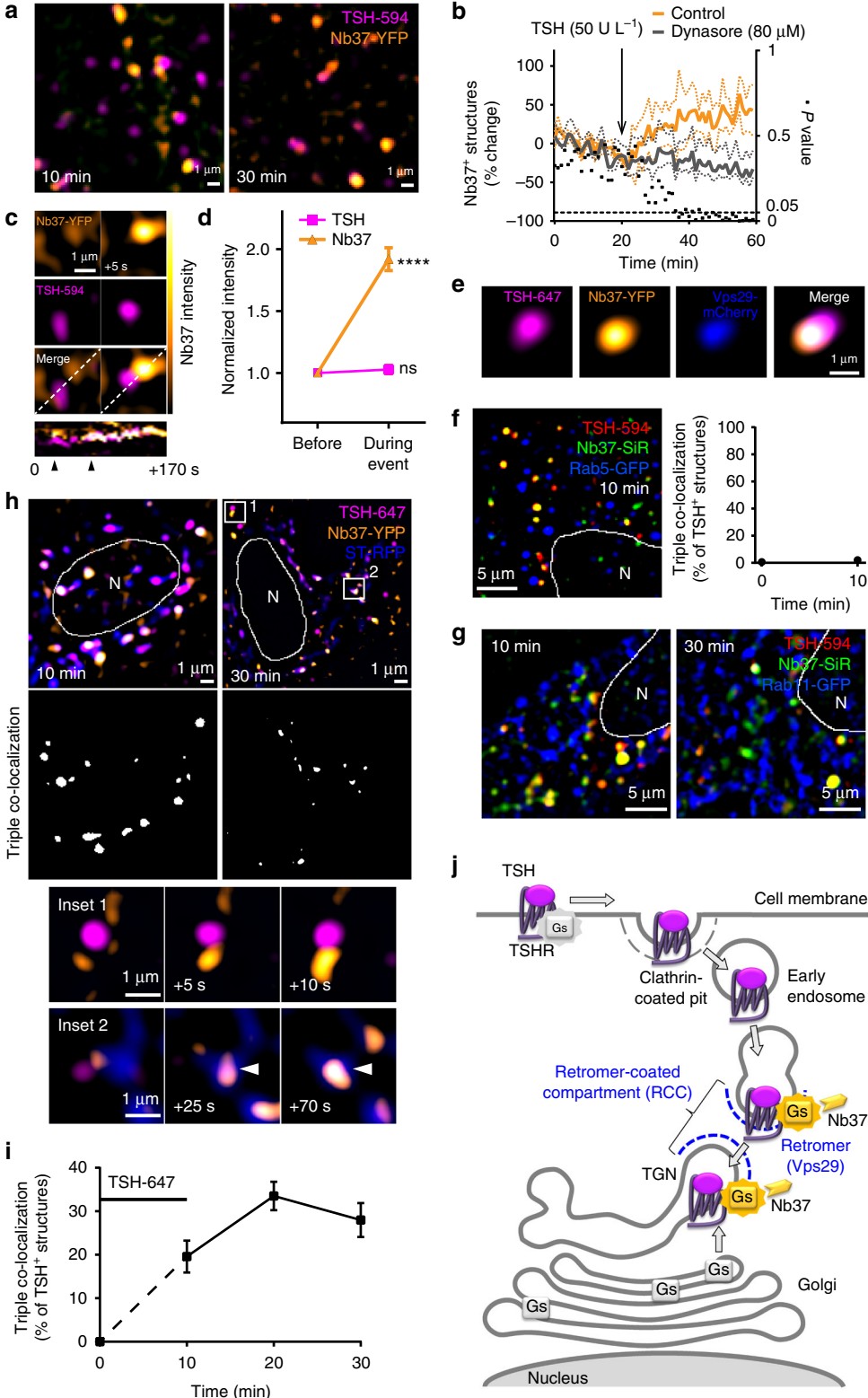

also in the remaining part. Trafficking among these structures typically occurred via fusion/splitting between two domains of the same type, i.e., prevalently carrying TSH/TSHR complexes or $G_s$-proteins (Fig. 2f and Supplementary Movie 3).

To better investigate the mechanisms involved in these trafficking events, we additionally visualized the retromer complex, which mediates retrograde trafficking of receptors and other membrane proteins from the endosome to the TGN[34–36].

To perform this task, the retromer complex segregates the membrane proteins targeted to the TGN on membrane subdomains of the endosome that eventually detach from it and fuse with the TGN[34–36]. For this purpose, primary mouse thyroid cells were cotransfected with $G_s$-YFP and fluorescently tagged retromer Vps29 subunit (Vps29-mCherry), followed by stimulation with fluorescently labeled TSH. We found that the subdomains where the internalized TSH/TSHR complexes were

co-localizing with the $G_s$-protein were characterized by the presence of Vps29 (Fig. 2g). Virtually all structures in which internalized TSH and the $G_s$-protein were simultaneously present were Vps29 positive (~96% of total; Fig. 2h).

Altogether, these findings indicated that the internalized TSH/TSHR complexes were selectively meeting the $G_s$-protein on membrane domains defined by the presence of retromer, which we call 'retromer-coated compartment' (Fig. 2i). The RCC comprises the retromer-coated domains found in the endosome and the TGN, which are in continuous dynamic exchange to mediate retrograde trafficking from the endosome to the TGN.

**Internalized TSHRs activate endogenous $G_s$-proteins at RCC.** To directly visualize activation of the endogenous $G_s$-protein, we used a conformational biosensor[15] based on a nanobody (Nb37) that selectively recognizes the guanine nucleotide-free state of the $G\alpha_s$ subunit[37,38]. This biosensor resides in the cytosol under resting conditions and translocates to membranes where $G_s$ activation occurs[15]. In these experiments, primary mouse thyroid cells were transfected with the biosensor (Nb37-YFP) alone to allow investigating the activation of endogenous $G_s$-proteins by endogenous TSHRs. In basal conditions, the biosensor (Nb37-YFP) was found to be cytosolic and partially located on the plasma membrane as well as on intracellular membranes, presumably because of basal $G_s$-protein activity (Supplementary Fig. 5; in subsequent images the cytosolic signal was suppressed to improve visualization). Upon stimulation with fluorescent TSH and receptor internalization, the TSH-containing perinuclear tubulovesicular structures recruited Nb37-YFP (Fig. 3a). This was observed as an increase (~60%) in the number of Nb37-YFP-positve structures over time (Fig. 3b). The effect was specific and required clathrin-mediated receptor endocytosis, as it could be prevented by pretreatment with the endocytosis inhibitor dynasore[39] (Fig. 3b), which efficiently blocked TSH/TSHR internalization in primary mouse thyroid cells (Supplementary Fig. 6). Moreover, we succeeded in directly observing an enhancement of the Nb37-YFP signal on $G_s$-protein-containing structures upon arrival of TSH-containing ones (Fig. 3c, d and Supplementary Movie 4)—note that on average the TSH intensity did not change during the same time, ruling out artefacts due to movements along the z-axis (Fig. 3d). In cells additionally transfected with Vps29-mCherry, recruitment of Nb37-YFP was observed on Vps29-postive subdomains (Fig. 3e), demonstrating that activation of $G_s$-protein was occurring in the RCC described in Fig. 2.

To further characterize the compartment where the internalized TSH/TSHR complexes activate the $G_s$-proteins, we simultaneously expressed Nb37-SNAP (stained with the cell-permeable benzylguanine derivative silicon-rhodamine, SiR) with the early endosome marker Rab5-GFP, the endocytic recycling compartment[40] marker Rab11-GFP, or the TGN marker ST-RFP[41]. Virtually no co-localization (i.e., <1% structures with triple co-localization) was observed with Rab5 (Fig. 3f) or Rab11 (Fig. 3g), indicating that there was no relevant activation of $G_s$-proteins on early endosomes or in the endocytic recycling compartment. Instead, we observed the simultaneous presence of internalized TSH/TSHR complexes and active $G_s$-protein on membrane domains that belonged to the TGN or were intimately associated with it (Fig. 3h, i; triple co-localization 34% at 20 min). Also for this triple co-localization, the kinetics was similar to that of TSH/TSHR accumulation in the TGN.

Taken together, the data presented in Figs. 1–3 indicated that $G_s$-protein activation induced by internalized TSHRs was occurring at a specific compartment defined by the presence of retromer, i.e., the RCC, which mediates retrograde trafficking to the TGN and is distinct from both early endosomes and the endocytic recycling compartment (Fig. 3j).

**PKA II in the Golgi/TGN promotes TSH nuclear signaling.** Based on these results, we set out to investigate the possible role of TSHR/$G_s$-protein signaling at the RCC in mediating the biological effects of TSH at cellular level. We concentrated our efforts on cAMP and its main effector PKA, which is a tetramer composed of two regulatory (R) and two catalytic (C) subunits[42]. The isoform of R subunit present plays a fundamental role in defining PKA specificity, by tethering the kinase to different subcellular compartments via interaction with specific scaffold proteins (AKAPs)[43].

First, we evaluated the subcellular localization of the main PKA subunits expressed in thyroid cells (C$\alpha$, RI$\alpha$, RII$\alpha$, RII$\beta$)[44] by immunofluorescence with specific antibodies and confocal microscopy (Fig. 4a). RII$\beta$, which plays a crucial role in mediating the nuclear effects of TSH[44] (see also below), was selectively localized on perinuclear tubulovesicular structures, consistent with its known location on membranes of the Golgi/TGN. RII$\alpha$ showed a prevalent localization on the same type of perinuclear tubulovesicular structures, which also contained a relevant fraction of C$\alpha$. Co-staining with a specific markers confirmed the selective localization of RII$\beta$ on membranes of the Golgi (Fig. 4b and Supplementary Fig. 7a), which is particularly extended in thyroid cells, and, to a lesser extent, of the TGN (Fig. 4c and Supplementary Fig. 7b). The pattern of RII$\beta$ and GOLPH4 immunofluorescence was clearly distinct from that of

**Fig. 3** Internalized TSH/TSHRs activate the $G_s$-protein at the retromer-coated compartment (RCC) associated with the TGN. Primary mouse thyroid cells were co-transfected with different combinations of Nb37-YFP (or -SNAP) and subcellular markers, followed by stimulation with fluorescent TSH (TSH-594 or TSH-647) for 10 min. **a** Recruitment of Nb37 to tubulovesicular structures containing internalized fluorescent TSH-594 at 10 and 30 min. **b** Quantification of the effect of TSH stimulation on the number of Nb37-positive (Nb37$^+$) structures over time. Reported is the percent change (mean ± s.e.m.; $n = 5/5$ cells) over basal with or without (control) dynasore pretreatment (30 min). Differences at individual time points were compared using the Holm–Šídák test for multiple comparisons (black dots, $P$ values). **c** Transient membrane recruitment of Nb37-YFP upon arrival of TSH-594-carrying vesicles. Top, Two consecutive images of a time series showing an increase of the Nb37-YFP signal upon arrival of a TSH-carrying vesicle. Bottom, Kymograph of the above time series generated along the indicated white line, showing two consecutive events of Nb37 recruitment (arrowheads). **d** Quantitative analysis of events like those shown in **c** (mean ± s.e.m.; $n = 30$). ****$P < 0.0001$ by Student's t-test compared to intensity immediately before the event. ns, statistically non-significant. Note that while the Nb37 intensity increased, the TSH-594 intensity did not change, ruling out artefacts due to movements along the z-axis. **e** Representative image of an Nb37-postive membranous structure, showing the asymmetrical localization of TSH-647, Nb37-YFP and Vps29-mCherry, used to visualize the retromer complex. **f, g** Simultaneous visualization of TSH-594, Nb37-SNAP labeled with SiR (Nb37-SiR) and Rab5-GFP or Rab11-GFP, showing lack of triple co-localization. **h** Simultaneous visualization of TSH-647, Nb37-YFP and the TGN marker ST-RFP. The insets show examples of Nb37 recruitment on both ST-RFP-positive and negative structures. **i** Quantification showing the percentage (mean ± s.e.m.; $n = 15$ cells) of TSH-containing structures co-localizing with Nb37-YFP and ST-RFP over time. **j** Schematic view of the results. Where appropriate, cell edges and nucleus (N) are marked in white. Data in **a**, **e**–**h** are representative of 4, 3, 3, 3 and 5 independent experiments, respectively

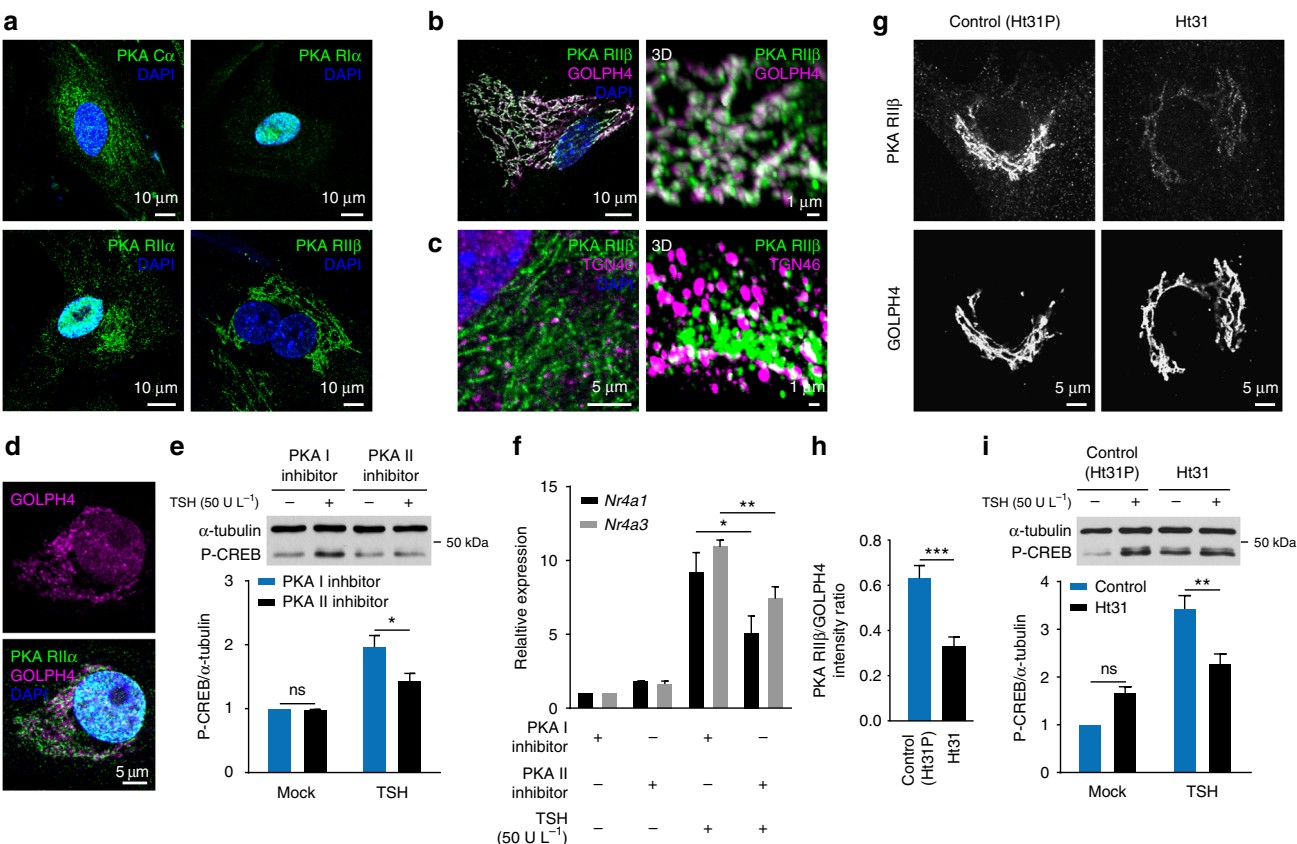

**Fig. 4** PKA II activation and localization at the Golgi/TGN are required for efficient TSH-dependent CREB phosphorylation **a** Subcellular localization of PKA subunits in primary mouse thyroid cells. The catalytic (Cα) and regulatory subunits (RIα, RIIα and RIIβ) of PKA were visualized by immunofluorescence with specific antibodies. Nuclei were stained with DAPI. **b**, **c** Co-staining of RIIβ with antibodies against GOLPH4 (**b**) and TGN46 (**c**), showing selective localization of RIIβ subunit on membranes of Golgi and TGN, respectively. Left, Confocal images. Right, 3D reconstructions from confocal image stacks (see Supplementary Fig. 7). **d** Co-staining of RIIα with GOLPH4, showing preferential localization of RIIα at the Golgi. **e** Western blot analysis of CREB phosphorylation (Ser133) in response to TSH in cells pretreated for 1 h with a PKA I (Rp8-Br-cAMPS, 250 μM) or PKA II (Rp8-Br-PIP-cAMPS, 250 μM) inhibitor. α-tubulin was used as loading control. Data (mean ± S.E.M, n = 3) are expressed relative to those in the mock-stimulated sample pretreated with PKA I inhibitor. **f** Induction of early genes in response to TSH in cells pretreated for 1 h with PKA I/II inhibitors. Data (mean ± S.E.M, n = 4) are expressed as in **e**. **g** Effect of Ht31 or Ht31P (control) peptides (20 μM, 1 h) on the Golgi localization of RIIβ. Shown are representative RIIβ and GOLPH4 immunofluorescence stainings acquired with same contrast settings. **h** Quantification of images as in **g**. RIIβ intensity values (mean ± s.e.m., n = 12/20) are expressed relative to those of GOLPH4. **i** Western blot analysis of CREB phosphorylation (Ser133) in cells pretreated for 1 h with Ht31 or Ht31P (20 μM) and stimulated with TSH. Data (mean ± s.e.m., n = 3) are expressed relative to those measured in the mock-stimulated control sample. Differences in **e**, **f** and **i** are statistically significant by two-way ANOVA. *$P < 0.05$, **$P < 0.01$, by Bonferroni's post hoc test. Difference in **h** is statistically significant by unpaired t-test (***$P < 0.001$). ns, statistically non-significant difference. Images in **a–d** and **g** are representative of 4, 3, 3, 5 and 3 independent experiments, respectively

the endoplasmic reticulum marker calnexin (Supplementary Fig. 8). A similar Golgi localization was found for RIIα (Fig. 4d).

We previously demonstrated in a well differentiated rat thyroid cell line (FRTL5) that PKA II, and in particular the isoform containing the RIIβ subunit (PKA-RIIβ), is required for the induction of CREB phosphorylation and gene transcription upon TSH stimulation[44]. To verify this critical role of PKA II in primary mouse thyroid cells, we repeated similar experiments in which we stimulated primary mouse thyroid cells with TSH in the presence of cell-permeable cAMP analogs that inhibit either PKA I or PKA II[44]. The results confirmed that activation of PKA II is required for the efficient induction of CREB phosphorylation (Fig. 4e). We also investigated the effect of PKA I/II inhibition on gene transcription of two typical cAMP/PKA regulated early genes in thyroid cells, i.e., nuclear receptor 4A1 (Nr4a1) and nuclear receptor 4A3 (Nr4a3), which are part of the initial transcriptional response to TSH that ultimately leads to increased thyroid hormone production and cell replication[44]. Inhibition of PKA II caused a significant reduction of the early gene response

to TSH as compared to PKA I inhibition (Fig. 4f). Furthermore, we interfered with the correct subcellular localization of PKA II by treating the cells with a cell-permeable peptide (Ht31) that inhibits RII binding to AKAPs[44]. Immunofluorescence experiments confirmed a partial loss of PKA RIIβ association with the Golgi in the presence of Ht31 as compared to control peptide Ht31P (Fig. 4g, h). This was accompanied by an inhibition of the TSH-dependent induction of CREB phosphorylation (Fig. 4i).

**TSHRs induce late cAMP/PKA response at Golgi/TGN**. We then used fluorescence resonance energy transfer (FRET) -based sensors to monitor in real time (by epifluorescence microscopy) the cAMP and PKA responses to TSH stimulation in primary mouse thyroid cells. At the end of each experiment, the cells were stimulated with forskolin to maximally activate the FRET sensors for normalization; the presence of a FRET response to forskolin indicated that neither sensor was saturated by TSH (Supplementary Fig. 9).

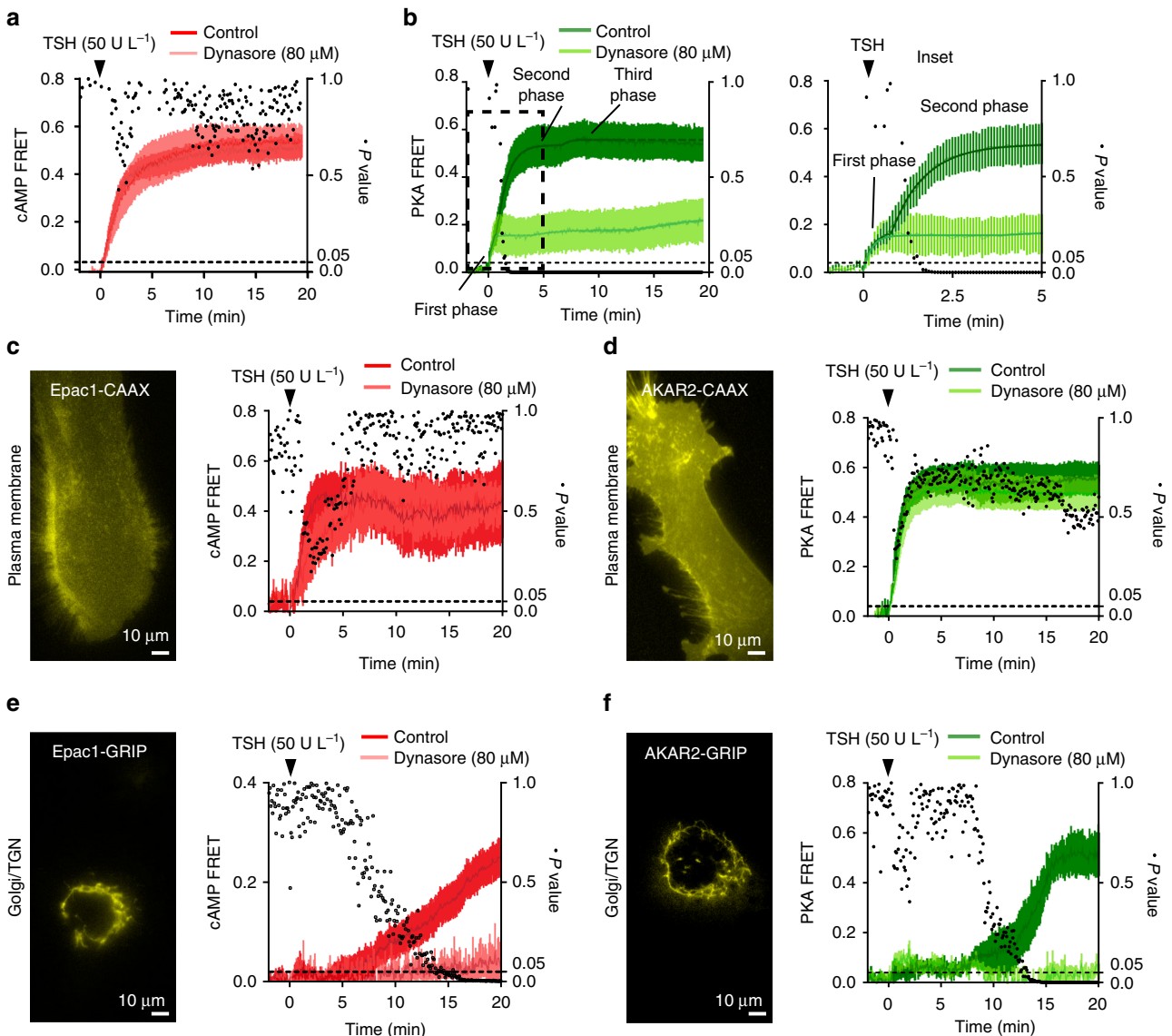

**Fig. 5** TSHR internalization is associated with a late cAMP/PKA response at the Golgi/TGN. Primary thyroid cells isolated from Epac1-camps transgenic mice or from wild-type mice and transfected with various FRET-based sensors were used to monitor cAMP and PKA responses in real time. Cells were pretreated with or without (control) dynasore for 30 min, followed by TSH stimulation. Data were normalized to the basal value (set to 0) and the maximal response to forskolin at the end of each experiment (set to 1). Differences at individual time points were compared using the Holm–Šídák test for multiple comparisons (black dots, *P* values). **a** cAMP FRET responses (mean ± s.e.m.; $n = 8/10$) in primary thyroid cells isolated from Epac1-camps mice. **b** PKA FRET responses (mean ± s.e.m.; $n = 11/7$) in primary thyroid cells transfected with the ubiquitous AKAR2 sensor. Data were fitted with a three phase exponential model with delay (see Methods). **c** cAMP FRET responses (mean ± s.e.m.; $n = 4/5$) in primary thyroid cells transfected with the Epac1-camps sensor targeted to the plasma membrane (Epac1-CAAX). **d** PKA FRET responses (mean ± s.e.m.; $n = 11/10$) in primary thyroid cells transfected with the AKAR2 sensor targeted to the plasma membrane (AKAR2-CAAX). **e** cAMP FRET responses (mean ± s.e.m.; $n = 8/4$) in primary thyroid cells transfected with the Epac1-camps sensor targeted to the Golgi/TGN (Epac1-GRIP). **f** PKA FRET responses (mean ± s.e.m.; $n = 5/4$) in primary thyroid cells transfected with the AKAR2 sensor targeted to the Golgi/TGN (AKAR2-GRIP)

First, we evaluated the kinetics of cAMP signaling in response to TSH stimulation. In primary thyroid cells isolated from a mouse transgenic for a cAMP FRET sensor (Epac1-camps)[13,45], the cAMP response to TSH was apparently characterized by a single phase (Fig. 5a, dark red), even though the kinetics was complex as the resulting curve could be better fitted with a three phase model than with a simple mono-exponential one (Supplementary Fig. 10a). Inhibiting TSH/TSHR internalization with dynasore did not cause any significant change in the overall cAMP response (Fig. 5a, light red); however, the resulting curve could be better fitted with a mono-exponential model than the control one (Supplementary Fig. 10b), consistent with a simpler

kinetics of cAMP accumulation. Moreover, the cAMP FRET response to forskolin was not affected by dynasore pretreatment, indicating that dynasore had no detrimental effects on cAMP production (Supplementary Fig. 11a).

Then, we monitored PKA activation by transfecting primary mouse thyroid cells with a FRET-based PKA sensor (AKAR2)[46]. The PKA response to TSH stimulation was characterized by a complex kinetics, with three consecutive phases discernible in both averaged (Fig. 5b, dark green) and individual traces (Supplementary Fig. 12a). In contrast to cAMP signaling, dynasore pretreatment greatly impaired the PKA response at late time points (>2 min). Whereas the first phase was conserved,

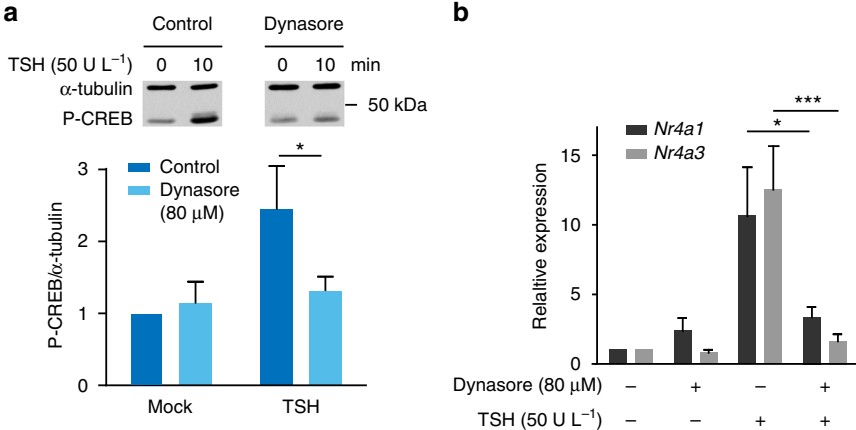

**Fig. 6** TSHR internalization is required for TSH-dependent induction of CREB phosphorylation and gene transcription. **a** Western blot analysis of CREB phosphorylation (Ser133) in primary mouse thyroid cells stimulated with TSH with/without dynasore pretreatment (30 min). α-tubulin was used as loading control. **b** Induction of early genes in response to TSH in primary thyroid cells with/without dynasore pretreatment (30 min). Data in **a** and **b** (mean ± s.e. m.; $n = 5$ each) are expressed relative to those measured in mock-stimulated control. Differences in **a** and **b** are statistically significant by two-way ANOVA. *$P < 0.05$ and ***$P < 0.001$ by Bonferroni's *post hoc* test

the second and third phases were largely reduced and/or delayed/slowed down (Fig. 5b, light green and Supplementary Table 1). At the same time, the PKA FRET response to forskolin was not affected by dynasore pretreatment, indicating that the effects of dynasore were specific for TSHR-induced signaling as they could be bypassed by directly activating adenylyl cyclases with forskolin (Supplementary Fig. 11b). These results suggested that while the receptors forced at the cell surface were still capable of inducing efficient cAMP generation, this was not sufficient to induce a normal PKA response.

We then sought to investigate where in the cell the different phases of cAMP/PKA signaling where occurring. To this aim, we generated Epac1-camps and AKAR2 sensors targeted to either the plasma membrane (Epac1/AKAR2-CAAX, Fig. 5c, d) or the Golgi/TGN (Epac1/AKAR2-GRIP, Fig. 5e, f), where PKA II is predominantly located. Experiments performed with the plasma membrane-tethered Epac1/AKAR2-CAAX sensors revealed that TSH stimulation was causing a rapid increase of cAMP levels and PKA activity at the plasma membrane that could not be prevented by dynasore pretreatment (Fig. 5c, d), consistent with the first phase observed with the global AKAR2 sensor. In contrast, experiments performed with the Golgi/TGN-tethered Epac1/AKAR2-GRIP sensors, showed that cAMP levels and PKA activity at the Golgi/TGN increased with a delay of ~7 and 9 min after TSH stimulation, respectively (Fig. 5e, f). These late responses at the Golgi/TGN were compatible with the kinetics of TSH/TSHR retrograde trafficking and Gs-protein activation at the TGN as well as with the third phase observed with the global AKAR2 sensor (Fig. 5b). Furthermore, the late Golgi/TGN responses could be prevented by dynasore pretreatment, suggesting that receptor internalization was required for cAMP/PKA signaling at the Golgi/TGN (Fig. 5e, f).

**TSHR internalization is required for TSH nuclear effects**. We then evaluated whether inhibiting TSHR internalization affected the PKA-dependent phosphorylation of CREB and gene transcription in response to TSH. Dynasore pretreatment prevented the induction of CREB phosphorylation in primary mouse thyroid cells (Fig. 6a). Consistently, dynasore pretreatment almost completely blocked the induction of early genes (Fig. 6b).

**TGN retrograde trafficking promotes TSH nuclear signaling**. We then sought to investigate the functional consequences of

TSHR-dependent cAMP/PKA signaling at the Golgi/TGN in thyroid cells. For this purpose, we first perturbed Golgi/TGN organization with Brefeldin A (BFA), which causes collapse of the Golgi into the endoplasmic reticulum (ER), impairs anterograde transport from the Golgi to the TGN and has been suggested to induce tubulation of the upstream endosomal compartment[47–49]. BFA pretreatment induced, as expected, disorganization of the TGN and reduced/delayed the co-localization of internalized TSH/TSHR complexes with the TGN marker ST-RFP, as shown by live-cell HILO imaging (Fig. 7a; for comparison see Fig. 1c). Immunofluorescence revealed that BFA treatment also caused dislocation of PKA RIIβ-subunit from the perinuclear region to small vesicles scattered throughout the cytoplasm (Fig. 7b). PKA RIIβ maintained some partial co-localization with TGN46, whereas the strong co-localization with the Golgi marker GOLPH4 was almost completely lost (Fig. 7b; for comparison see Fig. 4b, c).

We then evaluated the effect of BFA on the cAMP and PKA responses to TSH stimulation, measured by real-time FRET microscopy. Pretreatment of primary mouse thyroid cells with BFA significantly attenuated the cAMP response at late time points (>4 min) (Fig. 7c, light red), possibly because TSH/TSHRs were partially trapped in an endosomal compartment devoid of Gs-protein (and/or adenylyl cyclases). This was accompanied by a partial reduction of the second and third phases of PKA activation and apparently by a delay of the third phase, while the first one was not affected (Fig. 7d, light green; representative individual trace in Supplementary Fig. 12b). In contrast, the cAMP and PKA FRET responses to forskolin were not affected by BFA pretreatment, indicating that the effects of BFA were specific for TSHR-induced signaling as they could be bypassed by directly activating adenylyl cyclases with forskolin (Supplementary Fig. 11c, d).

Remarkably, BFA pretreatment almost completely prevented the TSH-dependent induction of CREB phosphorylation (Fig. 7e), similar to dynasore (Fig. 6a), suggesting that perturbing the Golgi/TGN organization uncoupled cAMP/PKA signaling from its nuclear effects.

In addition, we searched for a more specific way of inhibiting the retrograde trafficking of TSH/TSHR complexes. For this purpose, we silenced the retromer subunit Vps35 using specific siRNAs (Fig. 7f). The resulting partial reduction of Vps35 protein levels (~65%) was accompanied by a significant reduction of CREB phosphorylation upon TSH stimulation (Fig. 7f).

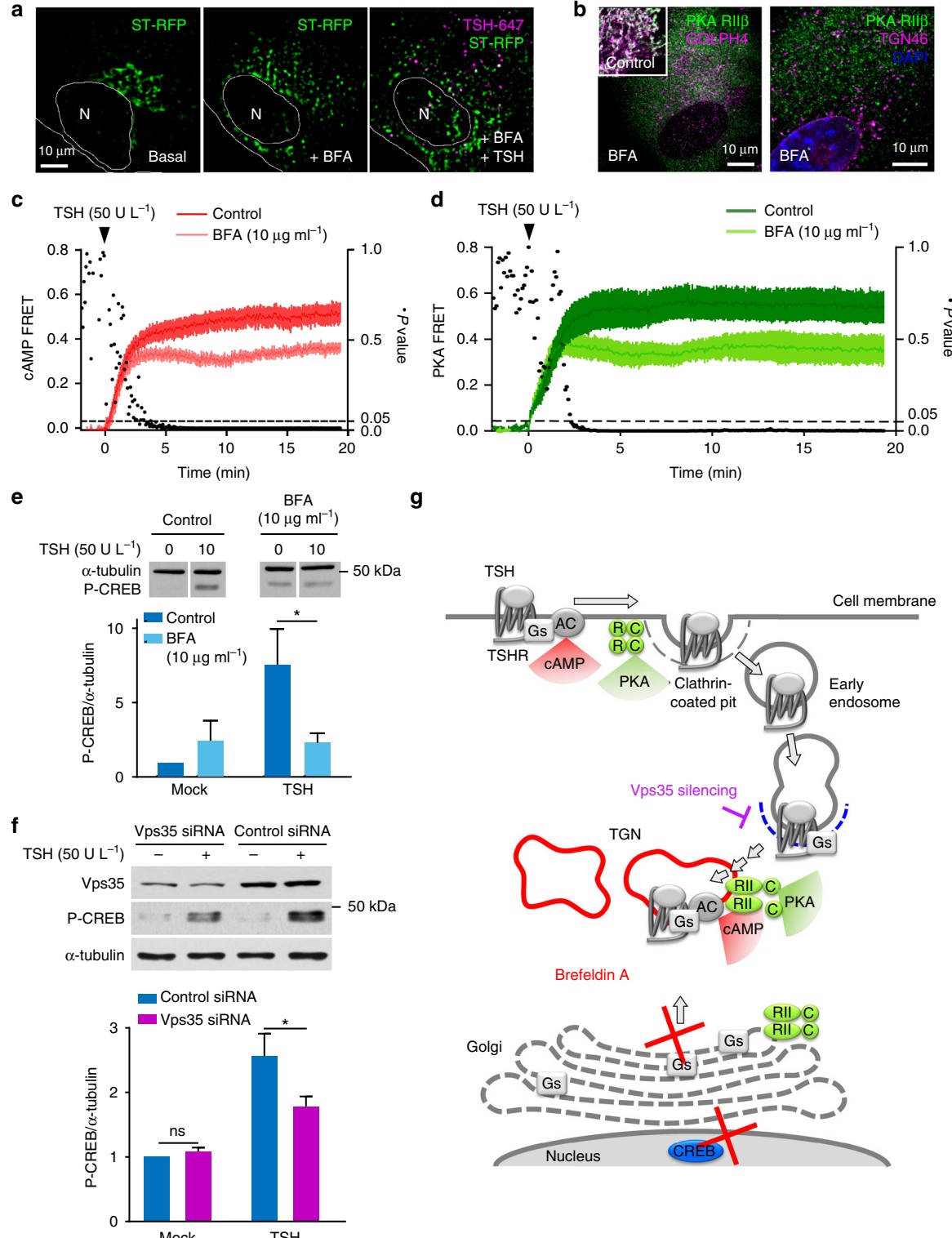

Altogether, these findings indicated that a correct organization of the Golgi/TGN as well as the retromer-mediated retrograde trafficking to the TGN were required for efficient CREB phosphorylation in response to TSH (Fig. 7g).

## Discussion

This study, which investigated endogenous TSHRs in primary thyroid cells, provides evidence that an agonist-activated GPCR can traffic retrogradely to the TGN and activate the $G_s$-protein in a retromer-coated compartment (RCC) that is both part of and intimately associated with the TGN, being responsible for retrograde trafficking from the endosome to the latter compartment[34,35]. Indeed, the fact that the TGN and in particular its retromer-coated subdomain is the only place where we simultaneously observe the presence of internalized TSH/TSHR, active $G_s$-protein and PKA II tends to exclude other compartments and suggests the TGN as the most likely critical site of signaling by internalized TSHRs.

Interestingly, we find that TSHR internalization is accompanied by a late phase of local cAMP/PKA signaling at the Golgi/TGN, with kinetics compatible with that of TSH/TSHR retrograde trafficking. Moreover, we show that interfering with receptor internalization, retrograde trafficking, PKA II activation/subcellular localization or Golgi/TGN organization all impair efficient CREB phosphorylation and/or gene transcription in response to TSH. On the basis of these findings, we propose a new model in which the internalized TSH/TSHR complexes traffic retrogradely to the TGN and induce local cAMP production and activation of PKA II—which is localized on membranes of the Golgi/TGN and is important for TSH nuclear effects—to ultimately induce efficient CREB phosphorylation in the nucleus and gene transcription (Fig. 8). This model is fundamentally different from the one put forward so far for PTH[14] and β₂-adrenergic[15] receptors, which have been shown to activate Gs-proteins on early endosomes and cease doing so upon entering the RCC[27,50]. Indeed, we observed no Gs-protein activation on early endosomes, a selective co-localization of internalized TSH/TSH complexes and Gs-protein in the RCC, and, just opposite to PTH receptors[27], a positive role of retromer in promoting CREB phosphorylation by internalized receptors. This indicates that the subcellular location and mechanisms involved in TSHR signaling after internalization are clearly distinct from those shown for PTH and β₂-adrenergic receptors.

The TGN plays an important role as a central hub for protein trafficking and sorting[34]. Interestingly, several key components of the GPCR machinery, including G-proteins and adenylyl cyclases, are abundant on intracellular membranes, above all of the Golgi/TGN[13,23]. While the presence of G-proteins on intracellular membranes has been general linked to receptor-independent roles in vesicular trafficking[23,51], recent data suggest a possible role of endosomal G-protein-mediated signaling in receptor trafficking[52]. PKA II is also predominantly tethered to membranes of the Golgi/TGN, from where it can translocate to the nucleus to regulate gene transcription[44,53]. We previously demonstrated that the expression of PKA II containing the RIIβ subunit (PKA-RIIβ) and its correct subcellular localization mediated by AKAPs are required for the TSH-dependent induction of CREB phosphorylation and gene transcription[44]. However, a direct link between GPCR signaling initiated at the plasma membrane and PKA signaling at the Golgi/TGN as well as a mechanism capable of explaining the requirement for such a high degree of spatial organization were lacking[54]. The present study provides an explanation for these findings by suggesting that it is the receptor (and note vice versa) that together with its ligand traffics to the TGN, where it activates resident Gs-proteins and induces local cAMP/PKA signaling to ultimately trigger nuclear effects. Altogether, these data point to the Golgi/TGN

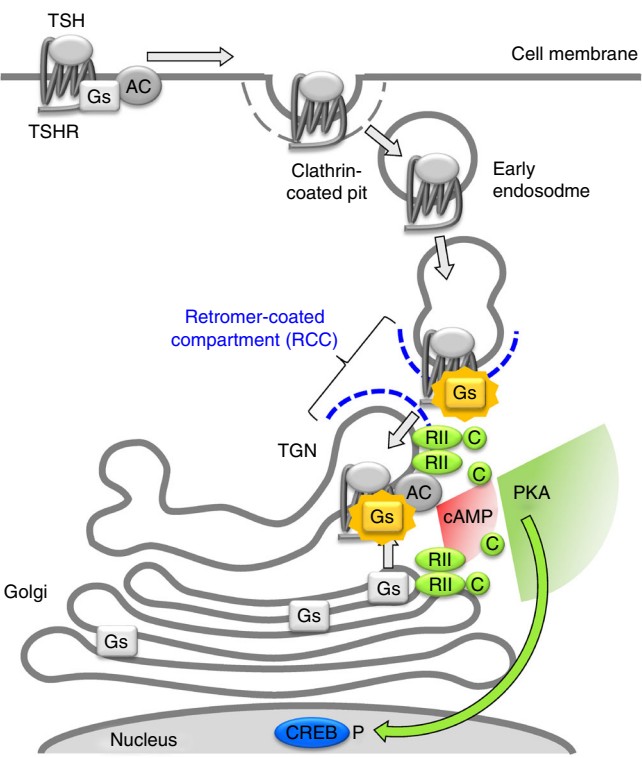

**Fig. 8** Model proposed for TSHR signaling at the TGN based on the results of this study. Upon TSH stimulation, TSH/TSHR complexes internalize and traffic retrogradely to the TGN, leading to local Gs-protein activation in the retromer-coated compartment (RCC) and cAMP/PKA signaling at the Golgi/TGN, which is critically positioned near the nucleus. This appears required for efficient PKA-dependent CREB phosphorylation and the induction of early genes in response to TSH, which in thyroid cells is known to ultimately lead to increased thyroid hormone production and cell proliferation

having a key role in GPCR action, by providing a dynamic platform that organizes all the components required for cAMP/PKA signaling at a critical location near the nucleus.

In addition to providing a mechanism capable of explaining the functional relevance of signaling by internalized TSHRs, the present findings strongly support the hypothesis of cAMP/PKA micro-compartmentalization[5,7–10]. Remarkably, we show that inhibiting receptor internalization with dynasore has no consequences on the overall amplitude of the cAMP response but prevents cAMP and PKA signaling at the Golgi/TGN as well as

**Fig. 7** Interfering with retrograde trafficking or Golgi/TGN organization impairs cAMP/PKA signaling and CREB phosphorylation. **a** Sequential HILO images showing effect of BFA on TGN organization and TSH/TSHR retrograde trafficking. Primary mouse thyroid cells were transfected with the TGN maker ST-RFP. Thereafter, they were treated with BFA for 30 min, stimulated with TSH-647 for 10 min and imaged for additional 30 min. **b** Immunofluorescence against RIIβ and either GOLPH4 (left) or TGN46 (right) in cells pretreated with BFA, showing displacement of the PKA RIIβ subunit from the Golgi and its reduced association with the TGN, respectively. Images were acquired by confocal microscopy. Data in **a** and **b** are representative of 3 independent experiments. **c** cAMP FRET responses (mean ± s.e.m.; n = 9/13) in primary thyroid cells isolated from Epac1-camps mice pretreated with or without (control) BFA for 30 min. **d** PKA FRET responses (mean ± s.e.m.; n = 11/11) in primary thyroid cells transfected with the AKAR2 sensor and pretreated with or without (control) BFA. Data in **c** and **d** were normalized as in Fig. 5. Differences at individual time points were compared using the Holm–Šídák test for multiple comparisons (black dots, P values). **e** Western blot analysis of CREB (Ser133) phosphorylation in primary thyroid cells stimulated with TSH with/without BFA pretreatment. α-tubulin was used as loading control. Images are representative of 3 independent experiments. Data (mean ± s.e.m., n = 3) are expressed relative to those measured in the mock-stimulated control sample. **f** Western blot analysis of CREB (Ser133) phosphorylation in primary thyroid cells transfected with Vps35 siRNA pool or control non-targeting siRNA and stimulated with TSH for 10 min. α-tubulin was used as loading control. Images are representative of six independent experiments. Data (mean ± s.e.m., n = 6) are expressed relative to those measured in the mock-stimulated control sample. Differences in **e** and **f** are statistically significant by two-way ANOVA. *P < 0.05 by Bonferroni's *post hoc* test. **g** Schematic view of the results

the induction of CREB phosphorylation and gene transcription. Moreover, altering the subcellular localization of PKA II via interfering with binding to AKAPs also impairs CREB phosphorylation. In addition, we show that interfering with the organization of the Golgi/TGN with BFA alters the kinetics of PKA activation and prevents CREB phosphorylation. Consistently, reducing retromer Vps35 subunit expression via siRNA partially impairs CREB phosphorylation in response to TSH. Taken together, these data suggest that what ultimately matters are not just the amplitudes of the cAMP and PKA signals but also their subcellular locations.

Another important finding of the present study is the direct observation of three kinetically distinct waves of PKA activation. This validates and further extends the idea of multiple waves of intracellular GPCR signaling that was previously only hypothesized[24]. On the basis of the results obtained with the plasma membrane and Golgi/TGN-targeted PKA sensors, we assign the first and third of these phases to PKA activation at the plasma membrane and at the Golgi/TGN, respectively. While we do not possess direct information about the second phase, a number of observations, including the fact that it precedes the Golgi/TGN phase and that the only other place where we observe the simultaneous presence of TSHR/$G_s$-protein and $G_s$-protein activation are retromer-coated domains associated with the TGN but lacking the TGN-marker ST-RFP, it is conceivable that the second phase occurs once the TSH/TSHR complexes enter the RCC but prior to their arrival in the TGN.

Whereas we believe that that this study represents a significant advance as it investigates endogenous receptors and $G_s$-proteins in their native cellular context and provides a new model to explain the functional consequences of G-protein-dependent signaling by internalized receptors at a cellular level, some limitations must be considered. First, we deliberately focused on early events within 30 min of TSH stimulation, i.e., induction of CREB phosphorylation and early gene transcription, and short pharmacological treatments (max. 1 h) to reduce possible indirect effects of interfering with receptor internalization/trafficking. This is a crucial issue that currently severely limits our possibilities of looking at long-term effects of GPCR signaling after internalization and/or to perform experiments *in vivo*. Second, despite the major advances represented by FRET and conformational biosensors, it remains difficult to directly localize the subcellular sites of cAMP production and PKA activation in living cells with high precision. Thus, the development of entirely new approaches and tools as well as of appropriate animal models appear required to investigate the physiological relevance of signaling by internalized receptors in the future.

Using a conformational biosensor for the $G_s$-protein, the present study was also able to capture the dynamic nature of $G_s$-protein signaling at intracellular sites. Our results indicate that $G_s$-protein activation at the RCC occurs in spikes with the arrival of new vesicles carrying TSH/TSHR complexes. This is reminiscent of the very transient events of $G_s$-protein activation observed for $\beta_2$-adrenergic receptors at early endosomes[15]. Altogether, these experiments reveal a quantal nature of GPCR signaling, whereby each vesicle carrying a defined amount of receptor and ligand seemingly produces a temporally and spatially restricted response.

Intriguingly, the present finding that TSHRs activate the $G_s$-protein in the RCC, whereas some other GPCRs do so on early endosomes[14,15,27,55]—and not in the RCC[50]—points to the existence of at least two distinct intracellular sites for signaling by internalized GPCRs. Both receptor characteristics and cellular context might ultimately dictate where and when a given receptor is signaling after internalization. In this regard, the modality of interaction with $\beta$-arrestin might also play an important role, as it

has been recently demonstrated that the vasopressin V2 but not the $\beta_2$-adrenergic receptor forms a stable supercomplex with the $G_s$-protein and $\beta$-arrestin, which favors endosomal signaling[56]. The requirement of TSHR internalization and retrograde trafficking to the TGN for efficient nuclear signaling and the existence of multiple intracellular sites for GPCR signaling offer new grounds to understand the cellular consequences of signaling by internalized receptors, as well as to explain receptor specificity, with important biological and pharmacological implications.

## Methods

**Animals.** Wild-type or transgenic FVB mice expressing the Epac1-camps sensor under control of the cytomegalovirus enhancer/chicken $\beta$-actin promoter[13] were used for the isolation of primary thyroid cells. All animal work was done according to regulations of the relevant authority, the government of Lower Franconia, Bavaria.

**Reagents and suppliers.** Cell culture reagents, collagenase I, collagenase II, goat serum, Alexa Fluor 594 succinimidyl ester, Alexa Fluor 647 succinimidyl ester, goat anti-rabbit Alexa Fluor 594-conjugated antibody (1:2,000), goat anti-mouse Alexa Fluor 647-conjugated antibody (1:2,000) and TRIzol reagent were from Thermo Fischer Scientific (Waltham, MA, USA). Dispase II, aprotinin, leupeptin and cOmplete MINI protease inhibitor cocktail tablets were from Roche Diagnostics (Mannheim, Germany). Trypsin/EDTA was from Pan Biotech (Aidenbach, Germany). Brefeldin A, dynasore, phenylmethylsulfonyl fluoride (PMSF), benzonase, bovine TSH, anti-PKA RIIβ rabbit polyclonal antibody (cat. no. HPA008421; 1:100) and anti-α-tubulin mouse monoclonal antibody (cat. no. T9026; 1:40,000) were from Sigma-Aldrich (Steinheim, Germany). Forskolin was from Tocris (Bristol, UK). Fetal bovine serum was from Biochrom (Berlin, Germany). ECL Prime Western Blotting Detection Reagent was from GE Healthcare (Buckinghamshire, UK). Mouse Accell SMARTpool Vps35 siRNA (cat. no. E-063309-00) and non-targeting control siRNA (cat. no. D-001910-01-20) were purchased from GE Healthcare Dharmacon (Lafayette, CO, USA). InCELLect AKAP St-Ht31 inhibitor peptide (cat. no. V821A) and St-Ht31P control peptide (cat. no. V822A) were from Promega (Madison, WI, USA). PKA I-selective inhibitor 8-bromoadenosine-3′,5′-cyclic monophosphorothioate, Rp-isomer (Rp-8-Br-cAMPS) and PKA II-selective inhibitor 8-piperidinoadenosine-3′,5′-cyclic monophosphorothioate, Rp-isomer (Rp-8-PIP-cAMPS) were from BIOLOG (Bremen, Germany). SNAP-Cell 647-SiR was purchased from New England BioLabs (Ipswitch, MA, USA). Rabbit polyclonal antibodies against GOLPH4 (cat. no. 28049; 1:1,000) and TGN46 (cat. no. 16059; 1:1,000) as well as a goat polyclonal antibody against Vps35 (cat. no. ab10099; 1:1,000) were from Abcam (Cambridge, UK). Anti-PKA Cα (cat. no. 610980; 1:100), RIα (cat. no. 610609; 1:100), RIIα (cat. no. 612242; 1:100) and RIIβ (cat. no. 610625; 1:100) mouse monoclonal antibodies were from BD Biosciences (Heidelberg, Germany). Anti-calnexin mouse monoclonal antibody (cat. no. NB300-518; 1:100) was from Novus Biologicals (Wiesbaden, Germany). Anti-phospho-CREB (Ser133) rabbit monoclonal antibody (cat. no. 4276; 1:1,000) was from Cell Signaling Technology (Danvers, MA, USA). Goat anti-mouse and anti-rabbit horseradish peroxidase (HRP) conjugated secondary antibodies (1:10,000) and goat anti-mouse Cy2-conjugated polyclonal antibody (1:400) were from Jackson ImmunoResearch (West Grove, PA, USA). All other chemicals were from AppliChem (Darmstadt, Germany). TSH fluorescently labeled with Alexa Fluor 594 (TSH-594) or Alexa Fluor 647 (TSH-647) was obtained as previously described[13].

**Plasmids.** A plasmid expressing the human wild-type TSHR tagged with EYFP (TSHR-YFP) has been previously described[57]. A plasmid expressing a fragment (37 amino acids) of α-2,6-sialyltransferase fused C-terminally to monomeric RFP (ST-RFP)[41] was kindly provided by Enrique Rodriguez-Boulan (Cornell University, New York, NY, USA). A plasmid expressing the $G\alpha_s$ subunit tagged with YFP ($G\alpha_s$-YFP) has been previously described[58]. A construct encoding His-tagged Nb37 was kindly provided by Jan Steyaert (VIB, Brussels, Belgium). Plasmids expressing Nb37-EYFP or Nb37-SNAP were generated by fusing EYFP or SNAP sequences, respectively, to the C-terminus of Nb37. Plasmids expressing Rab5 tagged with GFP (Rab5-GFP) and mCherry (Rab5-mCherry) were kindly provided by Tom Kirchhausen (Harvard Medical School, Boston, USA). A plasmid expressing EGFP-tagged Rab11[59] was kindly provided by Marino Zerial (Max Planck Institute of Molecular Cell Biology and Genetics, Dresden, Germany). A plasmid expressing YFP-tagged Vps29 (Vps29-YFP)[60] was a gift from Juan Bonifacino (NICHD, Bethesda, MA, USA). A plasmid expressing Vps29-mCherry was generated by replacing YFP with mCherry in the original construct. A plasmid expressing the plasma-membrane-targeted Epac1-CAAX sensor[61] was kindly provided by Viacheslav O. Nikolaev (University Medical Center Hamburg-Eppendorf, Hamburg, Germany). Plasmids expressing the AKAR2[46] and (cell membrane-targeted) AKAR4-CAAX[62] sensors were kindly provided by Jin Zhang (UCSD, San Diego, CA, USA). A plasmid expressing the plasma membrane-targeted AKAR2-CAAX sensor was generated by fusing the CAAX domain obtained from the latter plasmid

to the C-terminus of AKAR2. Plasmid pCMV6-KL5-GolginA1, expressing the Golgi/TGN-targeting GRIP domain of Golgin A1, was obtained from Origene. A plasmid expressing the Golgi/TGN-targeted AKAR2-GRIP sensor was generated by fusing the GRIP domain obtained from pCMV6-KL5-GolginA1 to the C-terminus of AKAR2. A plasmid expressing the Golgi/TGN-targeted Epac1-GRIP sensor was generated by replacing the AKAR2 sequence in the previous plasmid with that of the Epac1-camps sensor.

**Isolation and culture of primary mouse thyroid cells.** Primary mouse thyroid cells were prepared as described before[13,29]. Briefly, 2–4-month-old mice were killed by cervical dislocation. The thyroid lobes were removed and suspended in a 1.5 ml tube containing Dulbecco's modified Eagle's medium (DMEM)/F12, supplemented with 100 U ml$^{-1}$ collagenase I, 100 U ml$^{-1}$ collagenase II and 1 U ml$^{-1}$ dispase II. Digestion was carried out for 90 min at 37 °C with gentle shaking every 15 min. Follicles were then washed three times with complete culture medium (DMEM/F12 supplemented with 20% fetal bovine serum, 1% penicillin and 1% streptomycin). Thereafter, they were plated onto 100-mm Petri dishes and maintained at 37 °C, 5% (vol/vol) CO$_2$. Cells were cultured for 7–12 days in complete culture medium before use. To investigate CREB phosphorylation and gene transcription, cells were starved in DMEM/F12 without fetal bovine serum for 48 h before stimulation.

**HEK293 cell culture and transfection.** HEK293 cells were from American Type Culture Collection (ATCC). Cells were mycoplasma negative. Cells were cultured in DMEM supplemented with 10% fetal bovine serum, 0.1 mg ml$^{-1}$ streptomycin and 100 U ml$^{-1}$ penicillin at 37 °C, 5% (vol/vol) CO$_2$. HEK293 cells were seeded at a density of $2.5 \times 10^5$ cells per well onto six-well plates containing 24-mm round glass coverslips and allowed to grow for 24 h, after which they were transfected with the Effectene transfection kit (Qiagen) according to the manufacturer's protocol.

**Electroporation of primary mouse thyroid cells.** Electroporation of primary mouse thyroid cells was performed as previously described[29]. Briefly, cells were resuspended in 100 µl Dulbecco's phosphate-buffered saline (DPBS) and transferred to a 0.4-cm electroporation cuvette (Bio-Rad; Munich, Germany). 40 µg DNA was then added to the cuvette and electroporation was performed using a Bio-Rad Gene Pulser II electroporation system with a capacitance extender set on 0.32 kV and 125 µF. 1 ml warm complete culture medium was added immediately after the electric discharge and the cells were seeded on 24-mm round glass coverslips in six-well plates filled with complete culture medium.

**SNAP labeling.** Nb37-SNAP expressed in primary thyroid cells was labeled by incubating the cells with 1 µM of a cell-permeable SNAP substrate (SNAP-Cell 647 SiR) for 20 min at 37 °C in complete culture medium supplemented with 1% (w/v) bovine serum albumin (BSA). Cells were then washed three times using complete culture medium with 5 min incubation between each wash.

**Live-cell HILO imaging.** 48 h post transfection, live-cell imaging was carried out on a TIRF microscope (Leica AM TIRF) equipped with an EMCCD camera (Cascade 512B, Roper Scientific; Tucson, AZ, USA) and solid-state lasers (488, 561, 594 and 635 nm). An oil-immersion objective (HCX PL APO 100 × /1.46) was used for HILO. Transfected cells grown on 24-mm coverslips were imaged in an imaging chamber containing complete culture medium without phenol red. The microscope was equipped with an incubator to maintain the temperature at 37 °C. A quadruple band filter set was used to achieve minimal delay (1 ms switching time) among channels. Exposure time was 50–100 ms for each channel. Frames of image sequences were acquired every 3 or 5 s as indicated.

For the visualization of TSH/TSHR complexes, cells were stimulated with fluorescently labeled TSH (6 µg ml$^{-1}$ TSH-594 or 12 µg ml$^{-1}$ TSH-647) for 10 min, followed by three rapid washes with complete culture medium without phenol red. Visualization was interrupted during this stimulation time, as the strong fluorescence of TSH in the medium prevented imaging.

Post-acquisition image analyses was carried out on ImageJ (https://imagej.nih.gov/ij/). In experiments involving fluorescent ligands, bleaching correction using the Histogram method in ImageJ was performed. When indicated, a Fast Fourier Transform bandpass filter was applied to the images to suppress the signal from the cytosol. To improve visualization, images shown in insets were resized by a factor of 5.0 using bicubic interpolation, without introducing any noticeable alteration when compared at low magnification. For analyses based on quantification of the number of vesicles, a threshold was applied to the images and the vesicles were automatically detected and counted using the Measure Particles function of ImageJ. To estimate the percentage of vesicles with co-localization of two or more markers, a binary mask was generated from the images acquired in the first channel and was used to calculate the percentage of vesicles that were also positive in the other channel(s). Calculations were performed in MS-Excel.

**TSH quantification.** Single-molecule microscopy of individual fluorescently labeled TSH molecules spotted on clean coverslips was used for calibration. The

analysis confirmed the presence of 1-2 fluorophores per TSH-594 molecule, consistent with the spectrophotometrically determined degree of labeling. We calculated an average intensity of 400 a.u. at 100 ms exposure (or 4 a.u. ms$^{-1}$ exposure) for each TSH-594 molecule. We used this calibration to calculate the number of TSH molecules contained in individual TSH-carrying vesicles from cells imaged by HILO using the same settings (except for lower exposure time to avoid saturation while maintaining linearity between the two measurements). In parallel, we estimated the average volume of the vesicles based on their Feret diameters in 2D (median volume = 0.3 µm$^3$). From these data, we calculated a median concentration of $8.6 \times 10^{-7}$ M.

**Real-time FRET microscopy.** Primary thyroid cells isolated from Epac1-camps transgenic mice were plated on 24-mm round glass coverslips and used 48 h later for FRET measurements. In experiments using other FRET sensors, 7–12 day old primary mouse thyroid cells isolated from wild-type FVB mice were electroporated with plasmids expressing the sensors and imaged 48 h later. The AKAR2 sensor[46] was preferred over the more recent AKAR4 version[62], because we found AKAR2 to perform better in primary mouse thyroid cells. Coverslips were mounted in an imaging chamber and the cells were maintained in imaging buffer (137 mM NaCl, 5.4 mM KCl, 2 mM CaCl$_2$, 1 mM MgCl$_2$, 10 mM HEPES, pH 7.3). Imaging was performed on an Axiovert 200 inverted microscope (Zeiss; Jena, Germany), equipped with an oil-immersion objective (plan-NEOFLUAR 63×/1.25), a 505 dcxr beam splitter (Visitron Systems; Puchheim, Germany), a high-speed polychromator system (Visitron Systems) and an iXon Ultra EMCCD camera (Andor; Belfast, UK). The experiments were carried out at 37 °C and were monitored with MetaFluor 7 software (Molecular Devices; Sunnyvale, CA, USA). Images were acquired every 5 s and FRET was calculated as the ratio between YFP emission at 535 nm and CFP emission at 480 nm with correction for spillover of CFP signal into the YFP channel and for direct excitation of YFP[63]. For measurements using the Epac1-camps sensor, in which FRET values are inversely related to cAMP levels, the YFP/CFP ratio was inverted to allow direct comparison with PKA measurements. cAMP and PKA FRET responses were normalized to the maximal response of either FRET sensor obtained by treating the cells with the direct adenylyl cyclase activator forskolin (10 µM) at the end of each experiment[21].

**Immunofluorescence.** Samples were fixed with 4% paraformaldehyde for 15 min at room temperature (RT) and permeabilized with 0.1% Triton X-100 in DPBS for 3 min at RT. Thereafter, the samples were blocked with 5% goat serum in DPBS for 1 h at RT and incubated with the indicated primary antibodies overnight at 4 °C. Samples were then incubated with appropriate secondary antibodies (Cy2-conjugated anti-mouse, Alexa Fluor 594-conjugated anti-rabbit or Alexa Fluor 647-conjugated anti-mouse) at RT for 2 h. Nuclei were stained with DAPI. Coverslips were either mounted in 85% glycerol or stored at 4 °C in DPBS until imaging. Confocal images were acquired on a TCS SP5 confocal microscope (Leica Microsystems; Wetzlar, Germany) using an oil-immersion objective (HCX PL APO 63x/1.4).

**Pharmacological inhibition of PKA I and II.** For inhibition of PKA I or II, primary mouse thyroid cells were preincubated with either a PKA I-selective (Rp-8-Br-cAMPS) or a PKA II-selective (Rp-8-PIP-cAMPS) inhibitor for 1 h followed by stimulation with TSH.

**Retromer siRNA silencing.** The retromer Vps35 subunit was silenced using a pool of cell-permeable siRNAs against mouse Vps35 (Mouse Accell SMARTpool). A non-targeting siRNA was used as control. Silencing was performed by incubating the cells for 5 days with 1 µM siRNA in DMEM/F12 without fetal bovine serum.

**Western blot analyses.** Cells were washed with DPBS and directly lysed with lysis buffer (5% sodium dodecyl sulphate, 125 mM Tris-HCl, pH 6.8, supplemented with 0.1 mM PMSF, 2 µg ml$^{-1}$ aprotinin, 0.5 mg ml$^{-1}$ leupeptin, cOmplete Mini protease inhibitor cocktail) maintained at 95 °C. The cells were then collected with a rubber scraper and transferred to a 1.5-ml tube. Benzonase (final concentration 1 U ml$^{-1}$) was then added to the cell lysates followed by incubation at 4 °C for 1 h on a horizontal shaker (maintained at 300 r.p.m.). The lysate was then sonicated for 30 s and centrifuged at 20,000 $g$ at 4 °C for 30 min. The supernatant was then transferred to a fresh 1.5-ml tube and stored at −80 °C until use.

Cell lysates were separated on 10% SDS polyacrylamide gels and transferred to polyvinylidenefluoride membranes. Membranes were blocked in Tris-buffered saline containing 1% Tween-20 and 5% BSA. For the analysis of CREB phosphorylation, membranes were incubated with anti-phospho-CREB antibody overnight at 4 °C, followed by incubation with anti-tubulin antibody for 1 h at RT. Membranes were finally incubated with anti-mouse and anti-rabbit HRP-conjugated secondary antibodies for 1 h at RT. For the analysis of Vps35 expression, membranes were incubated with anti-Vps35 antibody overnight at 4 °C, followed by incubation with anti-goat HRP-conjugated secondary antibody for 1 h at RT. Membranes were treated with the ECL Prime Western Blotting Detection Reagent for 1 min at RT and detection was performed on photographic films (FUJI Medical; Tokyo, Japan). Films were scanned at 1,200 dpi and

densitometry analysis was performed using ImageJ. Uncropped immunoblot images can be found in Supplementary Fig. 13.

**Analysis of gene transcription by real-time PCR**. Real-time PCR was performed as previously described[44]. Briefly, RNA was extracted by the acid guanidinium thiocyanate-phenol-chloroform extraction method using the TRIzol reagent. RNA concentration was measured on a Nanodrop 2000 spectrophotometer (Thermo Fischer Scientific).

Single-stranded cDNA was synthesized using the SuperScript II Reverse Transcriptase kit (Thermo Fisher Scientific), starting from 1 μg RNA for each sample. The complementary RNA was then removed by adding 2 U RNase H per sample and incubating for 20 min at 37 °C.

Quantitative PCR was performed with the SYBR Select Master Mix (Thermo Fischer Scientific) on a C100 Real Time PCR system (Bio-Rad). Samples were measured in triplicate and were normalized against β-actin expression. The ΔΔCT method was used to calculate relative expression levels.

**Statistics**. Statistical analyses were performed using the Prism 6 software (GraphPad Software, La Jolla, CA, USA). Values are given as mean ± s.e.m. Differences between two groups were assessed by two-tailed Student's $t$-test. Differences among three or more groups were assessed by one-way or two-way analysis of variance (ANOVA), as appropriate, followed by Bonferroni's post hoc test. Differences between individual time points were evaluated using the Holm-Šídák test for multiple comparisons. Differences were considered significant for $P$ values < 0.05.

Data in Fig. 5b were fitted using the following set of equations:

$$Y(t) = A_1\left(1 - e^{-\frac{t}{\tau_1}}\right); \quad t \leq \Delta t_2$$

$$Y(t) = A_1\left(1 - e^{-\frac{t}{\tau_1}}\right) + A_2\left(1 - e^{-\frac{t-\Delta t_2}{\tau_2}}\right); \quad \Delta t_2 < t < \Delta t_3$$

$$Y(t) = A_1\left(1 - e^{-\frac{t}{\tau_1}}\right) + A_2\left(1 - e^{-\frac{t-\Delta t_2}{\tau_2}}\right) + A_3\left(1 - e^{-\frac{t-\Delta t_3}{\tau_3}}\right); \quad t \geq \Delta t_3$$

where $A_i$ is the amplitude of each phase $i$ (with $i = 1, 2, 3$), $\tau_i$ is the corresponding time constant and $\Delta t_i$ is the corresponding delay.

**Data availability**. The data that support the findings of this study are available from the corresponding author upon reasonable request.

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

## Acknowledgements

We thank Emanuele Cocucci for helpful discussions, Barbara Leibbrandt, Alexandra Bohl and Bianca Klüpfel for technical assistance, Ulrike Zabel for help with cloning and Martina Leo for help with the experiments using targeted cAMP/PKA sensors. This study was supported by the Deutsche Forschungsgemeinschaft (Grant CA 1014/1-1 to D.C. and Sonderforschungsbereich/Transregio 166–Project C1 to D.C.). A.G. was supported by a grant of the German Excellence Initiative to the Graduate School of Life Sciences, University of Würzburg.

## Author contributions

D.C. conceived the study. A.G. and S.L. performed the experiments and analyzed the data. A.G. and D.C. wrote the manuscript. M.J.L. discussed the results and edited the manuscript. D.C. supervised the study.

## Additional information

**Competing interests:** The authors declare no competing financial interests.

