## [Peer Review File · Nature Communications]

Reviewers' Comments:

Reviewer #1 (Remarks to the Author):

In this manuscript Godbole et al. extend recent reports showing canonical G protein signaling mediated by agonist-bound receptors in intracellular structures. The most significant new claim made by the authors is that active receptors travel as far as the TGN, and from this location (and this location only) they are capable of activating a pool of PKA that is in turn capable of phosphorylating CREB. The approach taken relies primarily on advanced live-cell imaging methods and FRET indicators of cAMP and PKA activity. While the results shown are generally consistent with the model laid out by the authors, they fall short of making a completely compelling case for a critical dependence on subcellular location. In part this is due to the longstanding difficulty of directly demonstrating spatial gradients of cAMP and PKA activity, which forces the authors to rely on indirect and correlative evidence for a location-dependent signal. In particular, the authors provide scant evidence that Gs is activated by TSHR in the TGN (Figure 2f, inset 2), and no direct evidence that this pool of Gs produces significant cAMP. Of course this does not mean that the authors' model is incorrect, but it does suggest that they will need to be more circumspect with respect to their conclusions.

Specifically, the authors should avoid drawing conclusions about the "position" of the PKA wave that is responsible for CREB activation, as they do not directly demonstrate a localized signal of any sort. For example, on line 250 the authors conclude that a PKA wave "triggered by internalized receptors at the TGN, is required for the induction of CREB phosphorylation". This is not supported by their data, which do not directly determine the source of the cAMP that activates the late PKA wave. The manipulations that inhibit CREB activation either dramatically impair bulk PKA activation (dynasore), or significantly inhibit it while disrupting the entire Golgi apparatus (brefeldin A). While the perinuclear location of the Golgi makes this an attractive hypothesis, there may be several other factors that make this an important organelle for CREB phosphorylation.

The authors use only two fairly crude pharmacological tools, dynasore and brefeldin A, to make the case that signaling at the TGN is critical for CREB phosphorylation. The manuscript would be strengthened considerably by the addition of supporting experiments with, for example, other receptors that produce similar cAMP signals but do not undergo internalization. Presumably these will be unable to stimulate CREB phosphorylation.

A critical missing control is to show that that fluorescent TSH is actually binding to TSH receptors, rather than binding nonspecifically to the cell surface. The experiment with TSHR-YFP is not really useful in this regard, as it only shows that the two labels can be found in the same structures. No experiments show that fluorescent TSH-positive structures actually bear native TSH/TSHR complexes.

One of the most striking observations in the manuscript is the odd separation of markers (TSH, NB37, ST-RFP) in what are interpreted as unitary membranous structures (vesicles and tubules). The authors make no attempt to explain how small vesicular structures can apparently “fuse” to allow TSH/TSHR complexes to encounter new Gs heterotrimers, while at the same time maintaining separation of the two membrane proteins in clearly distinct domains. How is Gs restricted to one half of the peanut? These are fascinating observations that warrant further investigation, but they do not fit cleanly into the models proposed by the authors.

The authors conclude that TSHR stimulates Gs in sorting endosomes, but this conclusion is based entirely on the appearance of some Nb37-positive structures and the fact that they are ST-RFP negative. Since no specific marker of sorting endosomes was used, this conclusion is not adequately supported.

The manuscript relies heavily in the first two figures on selected images showing colocalization, but not enough quantitative analysis is performed to support these images. The large majority of TSH-positive structures do not contain Gs-YFP, for example, and vice versa, yet the two markers are said to be colocalized.

The authors also should attempt to explain the differing kinetics of PKA activation, which appears to be maximal at about 2.5 minutes, and arrival of TSH at the TGN, which has barely begun at this time point.

Nb37-YFP experiments were presumably done with native (rather than overexpressed) Gs, but this is not made clear.

Nb37-YFP-positive structures are obviously quite mobile, therefore changes in intensity such as those shown in Figure 2 could easily be due to changes in the vesicle position relative to the focal plane. The authors should address this potential artifact.

Reviewer #2 (Remarks to the Author):

This manuscript has an interesting appeal, but only reports an incremental study in the evaluation

of TSH-receptor (TSHR) signaling via PKA in the trans-Golgi network that does not “explain the biological relevance of GPCR signaling at intracellular sites...” as concluded by the authors. The evidence for the existence of a TSHR recruiting nanobodies (Nb) for the active state Gs (Nb-37), and activating PKA in the TGN of thyroid cells is strong, but the physiological aspect of this TSHR signaling in the Golgi to bring biological relevance is not proven by the data or approach used here. There are also additional issues of concern with the signaling data that need to be addressed. Specific comments follow.

- The (over)use of “GPCRs” in the title/abstract is misleading giving that only the TSHR has been investigated (minor concern).

- In experiments using dynasore (Figures 1b, 3c, 3d, 3e, 3f), how does one know that dynasore can fully block TSHR internalization? Clearly, additional controls are needed to evaluate the efficacy of dynasore in preventing receptor endocytosis.

- The data of Fig. 3c are worrisome. While the two waves of PKA activation can be detected in Figure 3d, it is unclear why corresponding phases for the cAMP response cannot be observed. With further regards to these cAMP data, why dynasore has no effect? These points are particularly important to validate authors’ conclusion.

- The authors have unfortunately ignored an important role of G α s in the regulation of post-endocytic sorting of GPCRs (Roscioglione et al., Nat Commun 5: 4556, 2014). Could it be that the recruitment of Nb37 in the Golgi in response to activation of TSHR activation might also reflect the trafficking function of G α s?

Reviewer #3 (Remarks to the Author):

This manuscript addresses the subcellular localization of downstream effectors of GPCR signaling. An emerging paradigm is that GPCRs can still signal after they have been internalized in contrast to the dogma of desensitization occurring following ligand engagement at the plasma membrane. The novelty of this manuscript is that it suggests that signaling following stimulation of cells with TSH, the TSHR reaches the TGN via retrograde transport and it is from this location that PKA and subsequent gene transcription is activated. This supports the model whereby proximity to the nucleus is required for effective transcriptional activation. It also provides evidence contrary to a published report that TSHR signaling is terminated by delivery to the TGN.

This manuscript uses primary mouse thyroid cells which is a significant strength as the GPCR

trafficking and signaling field relies too heavily on the use of stable HEK cell lines (also used in previous study by Feinstein and colleagues) which may not accurately represent the relationship of trafficking and signaling of individual GPCRs in vivo. If it were the case that TSH receptor required trafficking to the Golgi to interact with active Gs proteins in order to activate transcription, this would be a novel and significant contribution to our understanding of TSH signaling. However there are a number of issues with the manuscript which undermine this interpretation of the data that are presented.

Specific issues:

The conclusion that delivery to the TGN is required for TSH signaling is based on two correlations: a. there are two kinetically distinguishable waves of PKA activation following internalization and b. activated Gs is detected at the TGN. The key issue is whether the delivery to the TGN is essential for gene transcription and it is difficult to be convinced that this is the case. An alternative is that Gs is activated at endosomes and continues to be active while being delivered to the TGN. The authors themselves point out that PKA activation can occur at sorting endosomes as well as the TGN. Furthermore it is difficult to resolve temporally what is happening during the signaling events. The second phase of PKA activation, supposedly occurring at the TGN, is initiated in a couple of minutes (Figure 3d) while the co-localisation of TSH with activated Gs is measured after 30 minutes. This begs the question as to how these events are related to each other. How quickly does activated receptor actually reach the TGN? Can the authors rule out the possibility that the delivery of the TSH and activated Gs occurs post TGN delivery rather than delivery being a requirement for it? The authors have attempted to address this question using BFA to disrupt the Golgi complex and although this prevents the appearance of activated CREB, it seems to merely delay PKA activation (Figure 4D). It is difficult to interpret this experiment and to eliminate effects that are independent of TGN disruption. For example can the authors eliminate that BFA does not tubulate sorting endosomes in these cells as has been reported for other cell lines (e.g. de Souza et al., *Curr Biol*, 2014)? This could also affect phosphorylation of Creb.

In order to demonstrate a requirement for TGN delivery a stronger approach would be to implicate more directly a requirement for the retromer sorting machinery in subsequent signaling. The use of siRNA targeting the retromer subunits or an acute depletion of retromer activity using the knocksideways approach is likely to give a more definitive answer.

We would like to thank the reviewers for expressing their interest in our study and for their valuable comments and suggestions.

Based on the reviewers' indications, we have performed a whole series of new experiments and analyses, which we believe address all their points and have allowed us to significantly improve our manuscript.

As you will see below, this includes the direct demonstration of local cAMP/PKA signaling at the Golgi/TGN using Golgi/TGN-targeted FRET sensors (answering the main point raised by Reviewer 1), the functional characterization of the PKA isoforms mediating TSH nuclear effects, as well as the demonstration that retromer silencing impairs the TSH-dependent induction of CREB phosphorylation (i.e. the main experiment suggested by Reviewer 3).

Altogether, we believe that our revised manuscript now makes a rather strong case that the TSH receptor traffics retrogradely to the TGN to induce local G_s-protein activation and local cAMP/PKA signaling, which appears critical for TSH-dependent nuclear effects. More specifically:

- We directly show that internalized TSH/TSH receptors selectively activate endogenous G_s-proteins at the retromer-coated compartment (RCC) that emanates from the TGN and mediates retrograde trafficking.
- We show that PKA II and its correct localization at the Golgi/TGN are required for TSH-dependent CREB phosphorylation in the nucleus.
- We show that receptor internalization and retrograde trafficking to the TGN are required for efficient CREB phosphorylation in response to TSH, which is ultimately responsible for increased thyroid hormone production and cell replication.

Our findings are clearly distinct from those obtained for other GPCRs (i.e. β_2 -adrenergic and PTH receptors), which have been shown to signal exclusively on early endosomes and to stop doing so upon entering the RCC, a crucial point that we feel was not sufficiently appreciated during our previous submission.

Moreover, we believe that our findings represent a significant step forward based on the following reasons:

- To the best of our knowledge, this is the first study that investigates the mechanisms of signaling by a GPCR at intracellular sites relying on endogenous receptors and endogenous G_s-proteins in their native cellular context (i.e. intact primary thyroid cells) – which is also indicated by Reviewer 3 as a major advance of our study.
- To the best of our knowledge, our findings demonstrate for the first time that an internalized GPCR activates the G_s-protein at the TGN, indicating the existence of multiple intracellular compartments for GPCR signaling (i.e. at least early endosomes and TGN).
- Our findings now also directly show local cAMP/PKA signaling at the Golgi/TGN and link retrograde TSH receptor trafficking to the nuclear effects of TSH – another major advance according to Reviewer 3.
- On the basis of our findings, we propose a completely new model to explain the mechanisms and functional consequences of signaling by internalized TSH receptors at a cellular level.

Point-by-point response to the reviewers' comments:

Reviewer #1 (Remarks to the Author):

In this manuscript Godbole et al. extend recent reports showing canonical G protein signaling mediated by agonist-bound receptors in intracellular structures. The most significant new claim made by the authors is that active receptors travel as far as the TGN, and from this location (and this location only) they are capable of activating a pool of PKA that is in turn capable of phosphorylating CREB. The approach taken relies primarily on advanced live-cell imaging methods and FRET indicators of cAMP and PKA activity. While the results shown are generally consistent with the model laid out by the authors, they fall short of making a completely compelling case for a critical dependence on subcellular location. In part this is due to the longstanding difficulty of directly demonstrating spatial gradients of cAMP and PKA activity, which forces the authors to rely on indirect and correlative evidence for a location-dependent signal. In particular, the authors provide scant evidence that Gs is activated by TSHR in the TGN (Figure 2f, inset 2), and no direct evidence that this pool of Gs produces significant cAMP. Of course this does not mean that the authors' model is incorrect, but it does suggest that they will need to be more circumspect with respect to their conclusions. Specifically, the authors should avoid drawing conclusions about the "position" of the PKA wave that is responsible for CREB activation, as they do not directly demonstrate a localized signal of any sort. For example, on line 250 the authors conclude that a PKA wave "triggered by internalized receptors at the TGN, is required for the induction of CREB phosphorylation". This is not supported by their data, which do not directly determine the source of the cAMP that activates the late PKA wave. The manipulations that inhibit CREB activation either dramatically impair bulk PKA activation (dynasore), or significantly inhibit it while disrupting the entire Golgi apparatus (brefeldin A). While the perinuclear location of the Golgi makes this an attractive hypothesis, there may be several other factors that make this an important organelle for CREB phosphorylation.

We thank the reviewer for recognizing the novelty and consistency of our study and for his/her suggestions. Based on the reviewer's comments we have generally modified the manuscript to be more circumspect with our conclusions. However, we think that our revised manuscript, which includes a whole new set of experiments and analyses, provides strong evidence for a critical dependency on subcellular localization. In particular, we do provide strong direct evidence that the Gs-protein gets activated by internalized TSHRs at a retromer-positive compartment associated with the TGN (see below). This has been achieved by means of a conformation-sensitive nanobody, which has been previously successfully used to directly show Gs-protein activation at early endosomes (Irannejad et al. Nature 2013). Moreover, there is a series of indirect but strong findings consistent with our model (see below). In addition, we have addressed the difficult question raised by the reviewer of directly visualizing the site of cAMP and PKA signaling (see below). Finally, we have performed a whole series of new experiments, including retromer silencing, which further support our model. Taken together, all these results are consistent with a model in which the internalized TSH/TSHRs traffic retrogradely to the TGN to induce local Gs-protein activation, which is accompanied by a delayed phase of cAMP/PKA signaling at the TGN and which appears required for efficient nuclear signaling in response to TSH.

Re. the question of directly localizing cAMP and PKA activation, we acknowledge that this would be an important improvement of our manuscript. At the same time, we agree with the reviewer on the extreme difficulty of performing such experiments, which is unfortunately a shared limitation of all the studies published so far on this topic. Nevertheless, we have taken up the challenge proposed by the reviewer and tried to directly visualize the location of cAMP production and PKA activation in thyroid cells stimulated with TSH.

A first hint that the site of cAMP production and PKA activation might change over time came from a derivative analysis of cAMP and PKA FRET data as we have described in a recent paper (Maiellaro et al. Cell Reports 2016 17:1238-1246). This analysis was partially already presented in **Supplementary Fig. 9** and we have repeated it with new images acquired with a higher signal to noise ratio (here shown only for review purposes):

Indeed, at least in some cells it was possible to see an initial phase when cAMP levels and PKA activity were increasing faster near the plasma membrane (PM), followed by at least a second phase in which the increase rate was higher in the perinuclear region.

To address this question more directly, we have generated cAMP and PKA sensors targeted to different cell compartments (i.e. the plasma membrane and the Golgi/TGN) and performed a whole new series of experiments and analyses (see revised **Fig. 5c-f**). In addition, we have increased the number of replicates in the experiments with the ubiquitous AKAR2 sensor (both under control conditions and in the presence of Brefeldin A; see revised **Fig. 5b** and **7d**) to better visualize the different phases and better evaluate possible differences. Remarkably, while we observed rapid activation of both plasma membrane-targeted sensors, the Golgi/TGN-targeted sensors were activated with a delay of approximately 7 and 9 min, respectively (**Fig. 5e,f**), compatible with the kinetics that we measured for TSH/TSHR retrograde trafficking to the TGN (**Fig. 1d**).

Fig. 5c-e. (see revised manuscript for details).

These experiments now allowed us to assign the first phase observed with the ubiquitous AKAR2 sensor to PKA activation at the plasma membrane and the third phase to PKA activation at the Golgi/TGN. Regarding the second phase, while we cannot draw direct conclusions, the available information suggests that it might occur when the TSH/TSHR complexes enter the retromer-coated compartment but prior to their arrival in the TGN (see Discussion, lines 358-367 for details).

Regarding the point raised by the reviewer that we provided “scant evidence that Gs is activated by TSHR in the TGN”, we do not agree on this specific point, as our findings directly show the colocalization of internalized TSH and active Gs-protein at the TGN and an enhancement of the active Gs-protein signal at the TGN upon arrival of TSH (see revised **Fig. 3g**). In the revised version of the manuscript we have improved the visualization of these results and added more quantitative analyses. Specifically, we have added a representative image at an earlier time point (10 min) (**Fig. 3g**) as well as colocalization images that help visualizing where the triple colocalization is occurring (**Fig. 3g**). Moreover, to make our case even stronger, we have added a quantification of the triple colocalization of Nb37/TSH/TGN marker transfected in primary thyroid cells (n=15 cells), showing that about 20% of the internalized TSH/TSHR complexes are found in the TGN together with the active Gs-protein as early as 10 min after stimulation and 35% at 20 min (**Fig. 3h**), which is similar to the kinetics of TSH accumulation in the TGN (**Fig. 1d**).

Furthermore, we have performed an additional quantification of the colocalization between internalized TSH, Gs protein and the retromer subunit Vps29. Importantly, this analysis shows that virtually all structures in which internalized TSH and the Gs-protein were simultaneously present were Vps29 positive (~96% of total). This indicates that the internalized TSH/TSH receptor complexes selectively meet the Gs-protein on membrane domains defined by the presence of retromer that are both part of

the TGN and in close dynamic association with it (see also response to point 2). The manuscript has been modified to include this important finding (see new **Fig. 2h** and Results, lines 155-161).

Re. the comment that “The manipulations that inhibit CREB activation either dramatically impair bulk PKA activation (dynasore), or significantly inhibit it while disrupting the entire Golgi apparatus (Brefeldin A).”, two new set of experiments that we have performed crucially address this point. First, our new experiments performed with the plasma-membrane-targeted cAMP/PKA sensors further demonstrate that the effect of dynasore is selective for signaling at intracellular sites, as plasma membrane signaling is not affected (**Fig. 5c,d**). Second, we have more selectively interfered with retrograde trafficking to the TGN via siRNA silencing of the retromer subunit Vps35 (**Fig. 7f**) (see point 8 below). The results of this experiment further support a role of retrograde trafficking to the TGN for efficient nuclear signaling in response to TSH. Here it is also important to mention that while dynasore dramatically affects PKA signaling, it does not impair bulk cAMP production, which we think provides an additional important control (see point 7 below for a detailed explanation). Moreover, we would like to point out that the forskolin-induced cAMP and PKA FRET responses were not affected by dynasore or Brefeldin A treatments (as now shown in the new **Supplementary Fig. 10**). Furthermore, we have deliberately decided to focus on early effects, since we are aware of the limitations of the currently available methods for inhibiting receptor internalization/trafficking, which might cause indirect or non-specific effect in case of long-term treatment - a point that we now explicitly mention in the discussion (see Discussion, lines 371-375).

In summary, we now believe that several direct and indirect findings of our study strongly support our model:

- 1) Differently from what observed for the β_2 -adrenergic and PTH receptors, we do not see colocalization between internalized TSH/TSHR and the Gs-protein on early endosomes, making receptor-dependent G-protein signaling on early endosomes rather unlikely.
- 2) We observe the simultaneous presence of internalized TSH/TSHR and either Gs-protein or Nb37 exclusively at the retromer-coated domains that emanate from the TGN. More precisely, we find these dynamic membrane domains both on endosomes (here we previously used the term “sorting” to underline the fact that these endosomes are the place where membrane proteins directed to the TGN are segregated/sorted in retromer-coated domains that transfer them to the TGN) and at the TGN. These domains are actually two sides of the same medal, as they mediate the retrograde transport from the endosome to the TGN and are in continuous dynamic exchange. Since this distinction between the retromer-coated domains at sorting endosomes and TGN is somewhat artificial and was apparently generating confusion, we have decided to collectively term these subdomains “retromer-coated compartment” (RCC).
- 3) The RCC is clearly distinct from early endosomes, again ruling out early endosomes as the site of intracellular signaling, which is in sharp contrast to what has been shown for the β_2 -adrenergic and PTH receptors.
- 4) Our new experiments with fluorescently labelled Rab11 (**see revised Fig. 3f**) also exclude an overlap between the RCC and the so-called “endocytic recycling compartment”.

5) PKA RII β is selectively localized and RII α is highly enriched on membranes of the Golgi/TGN in thyroid cells (**Fig. 4a-c**). C α also shows an enrichment at the Golgi/TGN (**Fig. 4a**). We could previously demonstrate in a well differentiated thyroid cell line (FRTL5) that PKA II (and not PKA I) mediates the effects of TSH on CREB phosphorylation and gene transcription (Calebiro et al. Mol. Endo. 2006 20:3196-3211). Moreover, in the same study we could show that this requires the presence and correct subcellular localization of RII β , which is in the Golgi/TGN. Importantly, in the revised manuscript (see **Fig. 4e,f**), we have repeated some of those key experiments in primary mouse thyroid cells. The new results confirm that PKA II mediates the effects of TSH on CREB phosphorylation (**Fig. 4e**) and that interfering with its subcellular localization (via Ht31, a cell-permeable peptide that interferes with RII binding to AKAPs) impairs the TSH-dependent induction of CREB phosphorylation (**Fig. 4f**). These results, together with those obtained with Brefeldin A, although indirect, make a strong case for the importance of activation of the PKA II pool located at the Golgi/TGN for nuclear signaling.

6) Our new experiments using targeted cAMP and PKA FRET sensors show that TSH stimulation causes a delayed phase of cAMP and PKA signaling at the TGN/Golgi (**Fig. 5e,f**), the kinetics of which is consistent with that of TSH/TSHR retrograde trafficking to the TGN (**Fig. 1d**).

7) The differential effect of dynasore on cAMP and PKA responses makes another strong case for a critical dependency on subcellular location. Specifically, this indicates that TSHR internalization is not required for efficient cAMP production (confirming our previous findings in intact thyroid follicles), presumably because there is a sufficient amount of ACs at the plasma membrane. However, consistent with the abundance of PKA at the Golgi/TGN, our data suggest that the receptor needs to induce signaling at or close to this intracellular PKA pool to induce a full PKA response and efficient nuclear signaling.

8) We have performed a new experiment in which we have silenced the retromer subunit Vps35 by siRNA (see **Fig. 7f**), as suggested by Reviewer 3. Importantly, the resulting partial reduction of Vps35 expression was accompanied by a significant reduction of CREB phosphorylation in response to TSH, further supporting the importance of retrograde transport to the TGN for nuclear signaling in response to TSH stimulation.

Indeed, the critical role of local signaling emerging from this study and the fact that the TGN and in particular its retromer-coated subdomains is the only place where we see the simultaneous presence of internalized TSH/TSHR, active Gs-protein (as well as total Gs-protein) and PKA II, tends to exclude other compartments and suggests the TGN as the most likely critical site of signaling by internalized TSHRs. In the revised manuscript, we have added the new experiments and analyses described above and have better clarified this fundamental point, including the strength and limitations of our approach (see Discussion lines 312-330 and lines 368-381).

The authors use only two fairly crude pharmacological tools, dynasore and brefeldin A, to make the case that signaling at the TGN is critical for CREB phosphorylation. The manuscript would be strengthened considerably by the addition of supporting experiments with, for example, other receptors that produce similar cAMP signals but do not undergo internalization. Presumably these will be unable to stimulate CREB phosphorylation.

We thank the reviewer for these suggestions. In the past ten years, we put a huge effort in trying to identify other GPCRs that induce cAMP signaling in primary mouse thyroid cells, largely motivated by the same idea of the reviewer. Despite some reports of other GPCRs being expressed in thyroid cells, we repeatedly failed in our attempts. For this reason, we are not able to perform this experiment.

However, we think that the comparison of cAMP and PKA signaling with/without dynasore provides a good internal control, as our new results with targeted cAMP/PKA sensors show that dynasore does not affect cAMP production and PKA activation at the plasma membrane (see new **Fig. 5c,d**) but only cAMP/PKA signaling at the Golgi/TGN (see new **Fig. 5e,f**).

Furthermore, the forskolin-induced cAMP and PKA responses were not affected by either dynasore or BFA treatment (see new **Supplementary Fig. 10**).

Finally, we have performed a new set of experiments, in which we have used a more selective way of interfering with receptor retrograde trafficking to the TGN. In these experiments, we have silenced the Vps35 retromer subunit and evaluated the effect on TSH-dependent CREB phosphorylation. Our results show that partial Vps35 silencing reduces the phospho-CREB induction in response to TSH (**Fig. 7f**), which is consistent with our model.

Fig. 7f. (see manuscript for details).

A critical missing control is to show that that fluorescent TSH is actually binding to TSH receptors, rather than binding nonspecifically to the cell surface. The experiment with TSHR-YFP is not really useful in this regard, as it only shows that the two labels can be found in the same structures. No experiments show that fluorescent TSH-positive structures actually bear native TSH/TSHR complexes.

We thank the reviewer for this comment. The fluorescently labeled TSH had been previously characterized in detail, including biological activity, specific binding to TSHRs expressed in HEK cells, competition by unlabeled TSH in primary thyroid cells, FRET between TSH and labeled TSHRs (see Calebiro et al. PLoS Biology 2009 7:e1000172 and Werthmann et al. 2012 26:2043-2048). In addition, we have performed new additional experiments in both HEK293 (see new **Supplementary Fig. 1**) and

primary thyroid cells (see new **Supplementary Fig. 2**) showing that fluorescent TSH binding (and internalization) is specific.

Regarding the presence of TSH/TSHR complexes in the endosome and TGN, we would like to point out that what we think is happening are dynamic interactions between receptors and TSH molecules. Still, given the high affinity of TSH for its receptor, TSH dissociation is rather slow and we hypothesize that most receptors are occupied by TSH in the intracellular structures where TSH and TSH receptors are both present. To provide support for this, we have performed a calibration with single-molecule imaging of individual TSH molecules and used it to estimate the number of TSH molecules present in TSH-containing vesicular structures. Based on this and on estimated volume of the vesicles, we estimated a TSH concentration in the lumen of the vesicles of approximately 800 nM. Since TSH has an affinity for its receptor in the picomolar range (high-affinity $K_d = 169$ pM), we deduced that nearly all receptors in these vesicles should have TSH bound to them. A paragraph and figure presenting these results have been added to the revised manuscript (see **Supplementary Fig. 4** and lines 122-125).

One of the most striking observations in the manuscript is the odd separation of markers (TSH, NB37, ST-RFP) in what are interpreted as unitary membranous structures (vesicles and tubules). The authors make no attempt to explain how small vesicular structures can apparently “fuse” to allow TSH/TSHR complexes to encounter new Gs heterotrimers, while at the same time maintaining separation of the two membrane proteins in clearly distinct domains. How is Gs restricted to one half of the peanut? These are fascinating observations that warrant further investigation, but they do not fit cleanly into the models proposed by the authors.

We agree with the reviewer that this is an interesting point that was not sufficiently explained in the original version of the manuscript. Actually, our results already contain an explanation for these findings, which has to do with the very way in which cargo is sorted out in the endosome and transported retrogradely to the TGN. The retromer complex plays an essential role in this process, by coating specific membrane domains of the endosome in which selected cargo, e.g. a receptor, accumulates and is segregated from the rest of the endosome even if the two domains are part of a unitary membranous structure. These retromer-coated structures eventually separate from the endosome and fuse with the TGN, still maintaining their retromer coating. Segregation in retromer-positive domains has been already shown for several receptors and other membrane proteins destined to the TGN (see e.g. references 34 to 36), and is believed to be the very way how sorting and retrograde transport actually work. Indeed, we think that it is probably the retromer machinery that keeps the Gs-protein excluded from the retromer-negative part of the endosome (possibly via intervention of SNX27 or another scaffold protein), while allowing TSH/TSHRs to enter the RCC and traffic to the TGN. While it would be interesting in the future to further dissect the involved molecular mechanisms, this goes beyond the scope of our current study. In the revised manuscript, we have better clarified these aspects (see lines 147-151).

The authors conclude that TSHR stimulates Gs in sorting endosomes, but this conclusion is based entirely on the appearance of some Nb37-positive structures and the fact that they are ST-RFP negative. Since no specific marker of sorting endosomes was used, this conclusion is not adequately supported.

We thank the reviewer for this comment, which underlines the need to bring more clarity on this important point. The main point that we want to make is that the TSHR activates the Gs-protein in the retromer-coated compartment (RCC) as defined in the first response to Reviewer 1 (see point 2) and that this is clearly distinct from early endosomes. The RCC mediates the retrograde trafficking of cargo from the endosome to the TGN. This occurs via retromer-coated domains that are assembled on subdomains of the endosome and then fuse with the TGN. In the previous version of the manuscript, we used the term “sorting endosomes” to define the endosomal structures that contained retromer-coated subdomains. The RCC is clearly distinct from early endosomes, which is a crucial point we want to make, as shown by the lack of colocalization with Rab5. Based on the reviewer’s suggestions, we have also performed additional experiments with fluorescently tagged Rab11 to further characterize this compartment. These experiments also exclude any involvement of the so-called “endocytic recycling compartment” (see **Fig. 3f**). Since we realized that the distinction between the endosomal part and TGN part of the RCC was somewhat artificial and was potentially generating confusion, we have decided to collectively name both parts “retromer-coated compartment” (RCC), as it is indeed the presence of retromer that rather precisely characterizes and defines this dynamic compartment (see also first response to Reviewer 1, point 2).

These findings are fundamentally different from what has been published for the β_2 -adrenergic and PTH receptor, which have been shown to activate Gs-proteins on early endosomes and to stop doing so after entering the retromer-coated domain (see Feinstein et al, Nat Chem Biol, 2011; Varandas et al Curr Biol 2016)

The manuscript relies heavily in the first two figures on selected images showing colocalization, but not enough quantitative analysis is performed to support these images. The large majority of TSH-positive structures do not contain Gs-YFP, for example, and vice versa, yet the two markers are said to be colocalized.

We thank the reviewer for these suggestions. The original version of the manuscript already contained a number of quantitative analyses, including a quantification of the colocalization between internalized TSH and the TGN marker ST-RFP over time (previous **Fig. 1b**), a quantification of the increase in the number of structures with active G protein (stained with Nb37) upon TSH stimulation over time with/without dynasore (previous **Fig. 2b**) and a quantification of the increase of Nb37 intensity upon arrival of new TSH (previous **Fig. 2d**). Moreover, based on the reviewer’s comment we have performed additional quantifications of our data (see below). Overall, the data reported in the revised manuscript is based on quantification of several cells from several independent experiments: **Fig. 1e** (n=14 cells at 4 different time points), **Fig. 2d** (n=10 cells at 4 different time points), **Fig. 3b** (n=5 time series), **Fig. 3d** (n=30 events), **Fig. 3h** (n=15 cells at 4 different time points).

More specifically, based on the suggestion of the reviewers we have performed additional experiments, added more images at early time points, added new colocalization images that help visualizing colocalizations, and a whole new series of quantitative analyses that are now shown in **Figures 1,2 and 3**.

These new analyses also directly address the point raised by the reviewer suggesting that “the large majority of TSH-positive structures do not contain Gs-YFP, for example, and vice versa, yet the two

markers are said to be colocalized". Indeed, our quantification (see **Fig. 2d**) shows that about 30% and 40% of the internalized TSH colocalized with the Gs-protein after 10 and 30 min, respectively, which we think is a relevant fraction. The manuscript has been modified to include the results of this and other quantifications.

Furthermore, we have performed an additional quantification of the colocalization between internalized TSH, Gs protein and the retromer subunit Vps29, which shows that virtually all structures in which internalized TSH and the Gs-protein were simultaneously present were Vps29 positive (~96% of total), which is now shown in **Fig. 2h** (see also first response to the reviewer's comments).

Fig. 2h. (see manuscript for details).

The authors also should attempt to explain the differing kinetics of PKA activation, which appears to be maximal at about 2.5 minutes, and arrival of TSH at the TGN, which has barely begun at this time point.

We thank the reviewer for raising this relevant question. Based on the reviewer's suggestions we have performed a whole series of new experiments and analyses to better investigate the kinetics of cAMP/PKA signaling, better characterize the previously identified phases and attempt to directly localize the sites of cAMP production and PKA activation (see revised **Fig. 5** and also first response to Reviewer 1). These included adding more replicates to the experiments with the ubiquitous AKAR2 sensor, in particular to better characterize the third phase and evaluate possible differences between control and Brefeldin A at late time points (see revised **Fig. 5b** and **7d**), which previously did not reach statistical significance. Moreover, we have generated plasma membrane- and Golgi/TGN-targeted Epac1/AKAR2 sensors and used them to directly compare the kinetics of cAMP/PKA signaling at the plasma membrane and the Golgi/TGN (**Fig. 5c-f**). Our results indicated that receptor internalization causes a delayed phase of cAMP/PKA signaling at the Golgi/TGN, with kinetics (delay approximately 7 and 9 min, respectively) that is compatible with that of TSH/TSHR retrograde trafficking to the TGN and with the third phase, which is now clearly visible in the averaged data obtained with the ubiquitous AKAR2 sensor. Experiments performed with the plasma membrane-targeted AKAR2 sensor clearly assigns the first phase to PKA signaling at the plasma membrane. Regarding the second, intermediate phase, although we cannot assign it directly, our indirect data suggest that it may occur once the receptor enters the RCC but before it reaches the TGN (see also first response to Reviewer 1).

Nb37-YFP experiments were presumably done with native (rather than overexpressed) Gs, but this is not made clear.

We thank the reviewer for this remark. Yes, the Nb37-YFP experiments were performed with endogenous Gs-protein, which we think is also an important advance compared to previous work. We have more explicitly stated this the revised version of the manuscript (see lines 28, 164, 167-169).

Nb37-YFP-positive structures are obviously quite mobile, therefore changes in intensity such as those shown in Figure 2 could easily be due to changes in the vesicle position relative to the focal plane. The authors should address this potential artifact.

We thank the reviewer for making this important point. These image sequences were acquired with a relative short frame interval (5s – and 3s for some). During this time, changes in z position were generally small. To exclude potential artefacts, we have additionally quantified the changes in TSH intensity over time in the same images, which we think provides a good internal control. The results, which have been added to current **Fig. 3d**, show that while the Nb37 intensity significantly increased (of approximately 80%), the TSH intensity did not significantly change.

Reviewer #2 (Remarks to the Author):

This manuscript has an interesting appeal, but only reports an incremental study in the evaluation of TSH-receptor (TSHR) signaling via PKA in the trans-Golgi network that does not “explain the biological relevance of GPCR signaling at intracellular sites...” as concluded by the authors. The evidence for the existence of a TSHR recruiting nanobodies (Nb) for the active state Gs (Nb-37), and activating PKA in the TGN of thyroid cells is strong, but the physiological aspect of this TSHR signaling in the Golgi to bring biological relevance is not proven by the data or approach used here. There are also additional issues of concern with the signaling data that need to be addressed. Specific comments follow.

We thank the reviewer for recognizing the potential interest and strength of our study and for his/her constructive criticisms and suggestions, which have helped us to improve the quality of our manuscript. As already stated in the responses to Reviewer 1, we have performed a series of new experiments and analyses, which we believe make an even stronger case that the TSHR traffics retrogradely to the TGN, where it activates a local pool of Gs-protein, leading to local cAMP and PKA signaling. This appears required for a fundamental cellular response, i.e. induction of CREB phosphorylation and gene transcription, which is well known to cause increased hormone production and proliferation in thyroid cells.

Here we would like to take once more the occasion to state that these findings are clearly different, and actually opposite, to what has been previously published for the β_2 -adrenergic and PTH receptors, which have been shown to signal exclusively at early endosomes and to stop doing so after entering the retromer-coated compartment (RCC). Moreover, although we previously hypothesized a possible role of

the TGN, our previous work (Calebiro et al. PLoS Biology 2009) only demonstrated that TSHR internalization was associated with persistent signaling, but did not demonstrate where in the cell this was happening. We also would like to stress that this study was performed on endogenous receptors and endogenous G proteins in primary cells, which as also stated by Reviewer 3, is a significant advance compared to previous studies in this field. Furthermore, we provide a new plausible mechanism (i.e. the vicinity of the Golgi/TGN to the nucleus) to explain the requirement of receptor signaling at intracellular sites for nuclear effects, which remains otherwise difficult to understand. Because of all these considerations, and because we identify a new compartment (i.e. the RCC that is associated with the TGN) where GPCRs are signaling – which is clearly distinct from the previously suggested early endosomes – we think that this study is indeed novel and represent a major step forward compared to the previous work of our and other groups.

Regarding the point raised by the reviewer that our study did not prove the “physiological” relevance of TSHR signaling at the TGN, we would like to stress that our study investigates the biological relevance at cellular level and not the physiological relevance of signaling by internalized TSHRs in vivo. In fact, we never used the term “physiological relevance”, as this would require studies in animals and the use of tools that are currently not available. Here, we deliberately focus on early events in cells to reduce possible indirect effects of interfering with receptor internalization/trafficking. Still, we perform this in intact primary cells at endogenous receptor and G protein levels, which has been a major effort and, as also stated by Reviewer 3, is a major advance compared to the previously published studies. The point we want to make is that signaling by internalized TSHRs is relevant for a key cell function, i.e. gene transcription (a key response to TSH stimulation, which ultimately leads to increased cell proliferation and thyroid hormone production) and that our study provides a plausible mechanism (activation of major Gs and PKA pools near the nucleus by internalized receptors) to explain it. We have revised our manuscript to better clarify these points and stress the limitations of currently available methods (see lines 270-272, and 368-381).

- The (over)use of “GPCRs” in the title/abstract is misleading giving that only the TSHR has been investigated (minor concern).

We thank the reviewer for this comment. In the revised manuscript we have replaced GPCRs with TSHR, as appropriate.

- In experiments using dynasore (Figures 1b, 3c, 3d, 3e, 3f,), how does one know that dynasore can fully block TSHR internalization? Clearly, additional controls are needed to evaluate the efficacy of dynasore in preventing receptor endocytosis.

We thank the reviewer for these suggestions. The fact that dynasore can efficiently inhibit TSHR internalization was already proven in our previous study (Calebiro et al. PLoS Biology 2009), consistent with what has also been shown by several groups on different GPCRs. To directly address the comment of the reviewer, we have performed new experiments and quantifications (now shown in **Supplementary Fig. 6**), which show that dynasore nearly completely blocks TSH/TSHR internalization in primary thyroid cells.

- The data of Fig. 3c are worrisome. While the two waves of PKA activation can be detected in Figure 3d, it is unclear why corresponding phases for the cAMP response cannot be observed. With further regards to these cAMP data, why dynasore has no effect? These points are particularly important to validate authors' conclusion.

We thank the reviewer for these comments. Based on these suggestions and those of Reviewer 1, we have performed a whole new series of experiments and analyses to better clarify the kinetics aspects of cAMP/PKA signaling, including the generation and use of plasma membrane- and Golgi/TGN-targeted cAMP and PKA FRET sensors (**Fig. 5**) (see also responses to the comments of Reviewer 1).

Re. the difference between cAMP and PKA kinetics, although three distinct phases can be clearly distinguished only with the PKA sensor, also the cAMP data can be poorly fitted with a simple exponential model and definitely better with the same model (3 phases with delay) used for the PKA data. Interestingly, however, the cAMP FRET data can be fitted much better with a single exponential model when internalization is inhibited with dynasore. Although we do not claim that we can extract quantitative information from these fittings as the phases are too overlapping for the fitting to be reliable, they clearly indicate that also the kinetic of cAMP production is complex (and consistent with our model, simpler in the presence of dynasore). These results are now shown in **Supplementary Fig. 9**.

Although several factors probably contribute to the differences between the cAMP and PKA data, here we would like to mention two factors that we think are particularly relevant:

First, cAMP is a small soluble molecule, whereas PKA is a tetrameric protein that is typically anchored to different intracellular structures via AKAPs. Thus, it is likely that whereas cAMP can diffuse on a longer distance and activate the Epac1 sensor at somewhat longer distances, the PKA effects remain more local. On this basis, one could expect the Epac1 responses to be less localized in time and space (and thus possibly underlying phases to be more "blurred") than those measured with the AKAR2 sensor.

Second, adenylyl cyclases and PKA seem to have a different overall subcellular localization, with a relevant fraction of PKA II being located on the Golgi/TGN. Our simplest interpretation of the fact that cAMP and PKA responses are differently affected by dynasore is that there is sufficient adenylyl cyclase at the plasma membrane to produce a maximal cAMP response, but not enough PKA to get a maximal PKA response. So what we think is happening is that while the TSHR internalizes, what changes is the site of cAMP production but not the bulk concentration of cAMP measured by the cytosolic Epac1 sensor. In contrast, the bulk PKA activity measured by the ubiquitous AKAR2 sensor increases as the TSHR accumulates in the TGN where a relevant fraction of PKA is located. We have better clarified this point in the revised version of the manuscript (see lines 244-248).

We believe that the lack of a major effect of dynasore on cAMP not only does not contradict our model, but, on the contrary, strongly supports it and provides an important additional control (see also first response to Reviewer 1, point 7 and second response to Reviewer 1).

- The authors have unfortunately ignored an important role of Gas in the regulation of post-endocytic sorting of GPCRs (Rosigligione et al., Nat Commun 5: 4556, 2014). Could it be that the recruitment of

Nb37 in the Golgi in response to activation of TSHR activation might also reflect the trafficking function of Gs?

We thank the reviewer for mentioning this. As was written in the previous version of the discussion, we are well aware of the studies showing a role of Gs in regulating trafficking events. While it is not possible to cite all those studies here, we definitely agree that the one by Rosciglione et al. might be particularly interesting in this context. We definitely do agree with the reviewer that besides being important for PKA activation and nuclear signaling, Gs-protein activation at the Golgi/TGN might also have consequences on (receptor) trafficking. Whereas this goes beyond the scope of the present study, it would be interesting to investigate also this aspect in the future. In the revised version of the manuscript we have more specifically mentioned this possibility and have added a reference to the study by Rosciglione et al. (see lines 334-335).

Reviewer #3 (Remarks to the Author):

This manuscript addresses the subcellular localization of downstream effectors of GPCR signaling. An emerging paradigm is that GPCRs can still signal after they have been internalized in contrast to the dogma of desensitization occurring following ligand engagement at the plasma membrane. The novelty of this manuscript is that it suggests that signaling following stimulation of cells with TSH, the TSHR reaches the TGN via retrograde transport and it is from this location that PKA and subsequent gene transcription is activated. This supports the model whereby proximity to the nucleus is required for effective transcriptional activation. It also provides evidence contrary to a published report that TSHR signaling is terminated by delivery to the TGN.

We thank the reviewer for recognizing the novelty of our study and for his/her comments and suggestions, which have greatly helped us to improve our manuscript. We are not aware of a previous study reporting that TSHR signaling is terminated by delivery to the TGN. The reviewer probably refers to the previous work by the group of JP Vilardaga on the PTH receptor (Feinstein et al, Nat Chem Biol 2011). Indeed, although our study and the previous one on the PTH receptor start from a similar observation, they come to diametrically opposite results and models for signaling by TSH and PTH receptors at intracellular sites: Signaling by PTH receptor occurs at early endosomes and is terminated once the receptor enter the retromer-coated compartment (RCC); Signaling by the TSHR does not occur on early endosomes and begins once the receptor enters in the RCC, which delivers it to the TGN.

This manuscript uses primary mouse thyroid cells which is a significant strength as the GPCR trafficking and signaling field relies too heavily on the use of stable HEK cell lines (also used in previous study by Feinstein and colleagues) which may not accurately represent the relationship of trafficking and signaling of individual GPCRs in vivo. If it were the case that TSH receptor required trafficking to the Golgi to interact with active Gs proteins in order to activate transcription, this would be a novel and significant contribution to our understanding of TSH signaling. However there are a number of issues with the manuscript which undermine this interpretation of the data that are presented.

We thank the reviewer for stating this important strength of our manuscript and for recognizing the potential significant contribution of our findings. Following his/her suggestions and those of the other reviewers, we have performed a whole new series of experiments and analyses, which we think address

all the points raised by the reviewers and make an even stronger case that while TSH/TSHR complexes traffic retrogradely to the TGN, they activate Gs-proteins at retromer-coated domains that are part of/intimately associated with the TGN, leading to local cAMP/PKA signaling near the nucleus, which appears required for efficient nuclear signaling in response to TSH (see below for details).

Specific issues:

The conclusion that delivery to the TGN is required for TSH signaling is based on two correlations: a. there are two kinetically distinguishable waves of PKA activation following internalization and b. activated Gs is detected at the TGN. The key issue is whether the delivery to the TGN is essential for gene transcription and it is difficult to be convinced that this is the case. An alternative is that Gs is activated at endosomes and continues to be active while being delivered to the TGN. The authors themselves point out that PKA activation can occur at sorting endosomes as well as the TGN.

We thank the reviewer for these comments. As detailed in the first response to Reviewer 1, we have performed a series of new experiments and analyses, including measurement of cAMP/PKA signaling at the Golgi/TGN via targeted FRET sensors (**Fig. 5e,f**), pharmacological activation of PKA I vs PKA II (**Fig. 4e**), displacement of PKA II with Ht31 (**Fig. 4f**) and, importantly, retromer (Vps35) silencing (**Fig. 7f**) as suggested by the reviewer (see below). Altogether, we believe that our new and previous findings make a rather strong case that the TSHR traffics retrogradely to the TGN, leading to Gs-protein activation at retromer coated domains that emanate from the TGN and inducing local cAMP/PKA signaling at the TGN. The new experiments performed by retromer silencing, showing a partial impairment of CREB phosphorylation after TSH stimulation (**Fig. 7f**), also further support the idea that TSHR signaling at the TGN is required for efficient CREB phosphorylation and gene transcription.

Based on the reviewer's comments we also realized that our previous distinction between sorting endosomes and TGN was misleading (see also first response to Reviewer 1, point 2). What we meant in reality is that receptor activation is happening on retromer-coated domains, which are responsible for retrograde transport from the endosome to the TGN. In fact, these are nothing but the two sides of the same medal, i.e. the retromer-coated domains of the endosome where receptors are sorted for delivery to the TGN (hence our use of "sorting endosome") and the retromer-coated domains at the TGN, where they are delivered. To avoid confusion, we have decided to collectively term these domains retromer-coated compartment (RCC). We agree with the reviewer that we cannot rule out that G proteins get at least partially activated in the RCC before reaching the TGN, but we think this doesn't change at all the conclusion of our manuscript and is actually only a rather semantic issue as the separation between the retromer-coated domains on the endosome and the TGN is only virtual. Indeed, these two "parts" are in rapid and continuous exchange in both directions, which is the very mechanism at the basis of retrograde cargo transport to the TGN.

Moreover, we have performed additional experiments to even better characterize this compartment, including colocalization experiments with Rab11 that excluded an involvement of "endocytic recycling compartment" (see new **Fig. 3f**).

The main point that we want to make here is that the RCC where we observe TSHR-dependent G protein activation is clearly distinct from early endosomes. Moreover, the Vps35 silencing experiments indicate that retromer is promoting, rather than inhibiting, nuclear signaling in response to TSH. This is

completely different and actually opposite to what has been published for the β_2 -adrenergic and PTH receptors, which have been shown to induce G protein signaling on early endosomes and to stop doing so once they leave the endosome and enter the retromer compartment.

In the revised manuscript we have better clarified this important point (see Discussion, lines 312-330).

Furthermore it is difficult to resolve temporally what is happening during the signaling events. The second phase of PKA activation, supposedly occurring at the TGN, is initiated in a couple of minutes (Figure 3d) while the co-localisation of TSH with activated Gs is measured after 30 minutes. This begs the question as to how these events are related to each other. How quickly does activated receptor actually reach the TGN? Can the authors rule out the possibility that the delivery of the TSH and activated Gs occurs post TGN delivery rather than delivery being a requirement for it?

We thank the reviewers for raising this point. Based on these comments and those of Reviewer 1, we have taken up the challenge of trying to better characterize the phases of PKA activation and the subcellular sites where cAMP and PKA signaling are occurring. For this purpose, we have generated cAMP and PKA sensors targeted to either the plasma membrane (Fig 5c,d) or the Golgi/TGN (Fig. 5e,f). The results allowed to even better identify three distinct phases of PKA activation, and clearly assign the first one to PKA signaling at the plasma membrane and the third one to signaling at the Golgi/TGN. Re. the second phase, our indirect data indicate that it might occur while the receptors enter the RCC but before they reach the TGN (see also response to Reviewer 1). We have also looked more carefully at the kinetics of TSH retrograde trafficking to the TGN and colocalization with Gs-protein as well as with Nb37, as a proxy of active Gs-protein. This included adding representative images at 10 min (see revised Figs. 1b, 1c, 2a, 2c, 2g, 3a, 3f, 3g) as well as more quantifications of colocalization over time (Figs. 2d and 3h). The previously and newly reported quantifications indicate that a relevant fraction (35%) of all internalized TSH is present in the TGN at 10 min and reaches almost 60% at 30 min, with similar kinetics for TSH/Gs-protein colocalization and TSH/Nb37/TGN marker colocalization. These kinetics are consistent with those observed for cAMP (Fig. 5e; delay of about 7 min) and PKA signaling (Fig. 5f; delay of about 9 min) at the Golgi/TGN, measured with the targeted sensors (consider that one might expect some delay between Gs protein activation and cAMP production as well as between cAMP production and PKA activation).

Moreover, we would like to mention that our new quantitative analysis shown in Fig. 2h shows that as high as 96% (!) of all the vesicular structures containing internalized TSH and Gs protein at 10 min are positive for retromer, clearly pointing to the RCC as the site of Gs-protein activation by internalized TSHRs.

The authors have attempted to address this question using BFA to disrupt the Golgi complex and although this prevents the appearance of activated CREB, it seems to merely delay PKA activation (Figure 4D). It is difficult to interpret this experiment and to eliminate effects that are independent of TGN disruption. For example can the authors eliminate that BFA does not tubulate sorting endosomes in these cells as has been reported for other cell lines (e.g. de Souza et al., Curr Biol, 2014)? This could also affect phosphorylation of Creb.

We thank the reviewer for raising this point. Based on the reviewer's suggestions and because there was already a trend (though not significant) for a decrease in the third phase of PKA activation in the presence of BFA, we have performed more experiments in the presence or absence of BFA to increase

the accuracy of our analysis. The extended data now show that BFA caused both a significant reduction and an apparent delay of the third phase (see **Fig. 7d**), which is now also more clearly visible in the control (**Figs. 5b** and **7d**). Overall, we think that these results are in rather good agreement with our model. Moreover, as suggested by the reviewer, we have performed a new set of experiments with retromer silencing (see response to next point), which provide a more specific way of interfering with retrograde trafficking.

Re. the issue of whether BFA also tubulates sorting endosomes, we agree that this could in principle also happen in our case (even if do not have evidence from our images), and might contribute to the effects of BFA. In the revised manuscript, we now mention this aspect and the work by D'Souza et al. Still, since our findings demonstrate that Gs-protein activation by internalized TSH receptors occurs specifically in the retromer-coated compartment (and not on early endosomes), that receptor internalization is required for cAMP/PKA signaling at the Golgi/TGN, that PKA II is largely localized in the Golgi/TGN and mediates the effects of TSH on CREB phosphorylation and that retromer silencing impairs efficient CREB phosphorylation in response to TSH, our data make a rather strong case that Gs-protein activation in the RCC/TGN and cAMP/PKA signaling at the Golgi/TGN is important for TSH-induced nuclear signaling.

Moreover, we would like to mention that we are well aware of the limitations of the currently available methods for interfering with receptor internalization/trafficking, which we now explicitly mentioned in the revised discussion (see Discussion, lines 368-381). This is the major reason why we have actually decided to focus on short term (cell) biological effects of TSH (i.e. induction of CREB phosphorylation and early gene transcription) and/or on short-term pharmacological treatments, as long-term treatments are more likely to be associated with indirect or non-specific effects.

For further details, see also the second response to Reviewer 1.

In order to demonstrate a requirement for TGN delivery a stronger approach would be to implicate more directly a requirement for the retromer sorting machinery in subsequent signaling. The use of siRNA targeting the retromer subunits or an acute depletion of retromer activity using the knocksideways approach is likely to give a more definitive answer.

We thank the reviewer for this highly valuable suggestion, which we have decided to follow. Although primary thyroid cells are very difficult to transfect, we succeeded in obtaining a good silencing of the retromer Vps35 subunit using cell-permeable siRNAs. The resulting partial reduction of Vps35 is associated with a significant reduction of CREB phosphorylation in response to TSH (**Fig. 7f**). We believe that this crucial result now makes an even stronger case for an important role of retrograde TSHR trafficking and signaling at the TGN for efficient nuclear signaling in response to TSH.

Note: In the revised manuscript, we have highlighted the major changes compared to the previous version in red. We would also like to mention that in the previous version of Table 1, k values were given instead of tau values by mistake. We apologize for this and have corrected it in the revised manuscript.

Reviewers' Comments:

Reviewer #1 (Remarks to the Author):

The authors have done a good job of addressing my comments and have added significant new results to the manuscript. Figure 5 in particular is important and quite nice. Well done.

Reviewer #3 (Remarks to the Author):

In the revised manuscript the authors have made significant efforts to address the reviewers' comments. Although the additional data are supportive of the authors' conclusions, some further experiments are required to provide a compelling case for TGN delivery of TSH and TSHR for effective signalling. Some of the additional experiments included in the manuscript would benefit from additional controls or have not been tested in the most compelling assays available to the authors.

In my initial review, my main concern was whether there is an absolute requirement for delivery to the TGN for TSH signalling or whether it occurs as a continuum starting in early/sorting endosomes. The authors have tried to address this by knocking down Vps35, which results in a partial reduction of CREB phosphorylation which could be explained by the partial knockdown of the protein. Although this supports a requirement for retromer for TSH signalling, it would be good to support these data by additional controls in some of the other figures. For example:

1. Based on the IF images presented, there are pools of PKA isoforms throughout the cell and indeed the authors' assertion that there is co-localisation with the Golgi is not always supported by the images presented. In Fig 4d, the staining of both Golgi 4 and PKA looks much more like ER staining, a result which contrasts with the images presented in Figure 4a. Given that different pools of AKAPs are required for membrane association of PKA, it would be important to show that the PKA inhibitors used in Fig4 (e and f) cause a specific loss of association of PKA from the TGN/retromer compartment.
2. A related point is whether these pharmacological inhibitors also affect gene transcription. The authors have shown that dynasore interferes with CREB-mediated gene transcription but since the point of the manuscript is to show that specific delivery to the TGN is required, it would be important to carry out the same experiments with the specific PKA inhibitors.
3. The argument that delivery to the retromer compartment is essential could be further strengthened by demonstrating that Nb37 does not co-localise with EEA1 or indeed rab5.

Minor points: The title of the manuscript refers to trafficking to the TGN whereas in the rebuttal the authors state:

‘We agree with the reviewer that we cannot rule out that G proteins get at least partially activated in the RCC before reaching the TGN, but we think this doesn’t change at all the conclusion of our manuscript and is actually only a rather semantic issue as the separation between the retromer-coated domains on the endosome and the TGN is only virtual.’

Actually, although apparently semantic, it is important to define as accurately as possible the nature of intracellular compartments that are involved in signalling. The reality is that the environment of the TGN is likely to be quite different from that of retromer coated domains on endosomes and this is important to bear in mind for any future mechanistic studies. Hence the title should reflect this distinction.

2. On a similar theme, I suggest that the authors might consider calling the second and third phases of PKA activation ‘later’ rather than ‘delayed’ which suggests a defect whereas in fact they are arguing that these later waves are the physiologically relevant ones.

Reviewers' comments:

Reviewer #1 (Remarks to the Author):

The authors have done a good job of addressing my comments and have added significant new results to the manuscript. Figure 5 in particular is important and quite nice. Well done.

We would like to thank once more the reviewer for the very useful comments and suggestions and we are very pleased that the reviewer was fully satisfied with our revisions.

Reviewer #3 (Remarks to the Author):

In the revised manuscript the authors have made significant efforts to address the reviewers' comments. Although the additional data are supportive of the authors' conclusions, some further experiments are required to provide a compelling case for TGN delivery of TSH and TSHR for effective signalling. Some of the additional experiments included in the manuscript would benefit from additional controls or have not been tested in the most compelling assays available to the authors.

We would like to thank the reviewer for acknowledging our previous efforts and the fact that our additional data were supportive of our conclusions. We also appreciate the additional comments and suggestions of the reviewer, which have helped us to further improve our manuscript.

Based on the reviewer's suggestions, we have performed additional experiments and controls, which we think make the main conclusions of our study even stronger (see below for details; main changes in the manuscript are marked in blue).

In my initial review, my main concern was whether there is an absolute requirement for delivery to the TGN for TSH signalling or whether it occurs as a continuum starting in early/sorting endosomes. The authors have tried to address this by knocking down Vps35, which results in a partial reduction of CREB phosphorylation which could be explained by the partial knockdown of the protein.

We agree that the retromer (Vps35) silencing experiment, which was one of the key experiments suggested by the reviewer during the first review, supports the requirement of retromer-mediated retrograde trafficking for efficient TSH signaling.

Our extensive experiments had already demonstrated the virtual lack of internalized TSH/(TSHR) together with Gs protein on early endosomes, whereas as high as 96% (!) of all the vesicular structures containing internalized TSH/(TSHR) and Gs protein are positive for retromer already at 10 min.

Moreover, based on the additional reviewer's suggestions we have performed further experiments and quantifications (see below), which show the virtual lack of activation of Gs protein on early endosomes.

Altogether, we believe that our previous and new results provide strong evidence that activation of Gs protein by internalized TSH receptors is not occurring in early endosomes, but only after the receptors enter the RCC, which is the main conclusion of our study.

Although this support a requirement for retromer for TSH signalling, it would be good to support these data by additional controls in some of the other figures. For example:

1. Based on the IF images presented, there are pools of PKA isoforms throughout the cell and indeed the authors assertion that there is co-localisation with the Golgi is not always supported by the images presented. In Fig 4d, the staining of both Golph 4 and PKA looks much more like ER staining, a result which contrast with the images presented in Figure 4a.

We thank the reviewer for this comment, which underlines the fact that the general reader might not be familiar with the very extended organization of the Golgi in thyroid cells. Both images in **Figure 4a** and **4d** are actually consistent with Golgi in thyroid cells, which is supported by several immunofluorescence (IF) and electron microscopy experiments we have performed in the last 10 years (see for instance Calebiro et al. PLoS Biology 2009 e1000172). We have better clarified this aspect in the newly revised manuscript (see lines 205-206).

Moreover, we have performed additional IF experiments with an endoplasmic reticulum (ER) marker calnexin to directly address the reviewer's question. The results reported in the **new Supplementary Figure 8** show that the pattern of RII β and GOLPH4 immunofluorescence is clearly distinct from that of the ER marker.

Given that different pools of AKAPs are required for membrane association of PKA, it would be important to show that the PKA inhibitors used in Fig4 (e and f) cause a specific loss of association of PKA from the TGN/retromer compartment.

The inhibitors used in previous **Figure 4e** are actually cAMP analogs that selectively inhibit PKA I or II. These inhibitors are not expected to cause any dissociation of PKA from the Golgi/TGN, and that has never been our claim.

In contrast, Ht31 used in previous **Figure 4f** (**now Fig. 4i**) interferes with the binding of PKA RII subunits to AKAPs and is expected to cause dissociation of PKA from the Golgi/TGN. Based on the reviewer's suggestion, we have performed additional experiments in which primary mouse thyroid cells were treated with Ht31 or control peptide (Ht31P) and analyzed by immunofluorescence. We focused on the Golgi because that's where the vast majority of RII β is localized and because we cannot reliably evaluate changes in other compartments that contain much less (i.e. TGN) or virtually no RII β . The results indicate that Ht31 treatment was associated with a partial loss of RII β at the Golgi compared to a Golgi-resident protein (GOLPH4), which is consistent with the partial reduction of CREB phosphorylation in response to TSH that we observe with Ht31 (see **Fig. 4i**). These new results have been included in the main text (see lines 222-224 and **Fig. 4g,h**).

2. A related point is whether these pharmacological inhibitors also affect gene transcription. The authors have shown that dynasore interferes with CREB-mediated gene transcription but since the point of the manuscript is to show that specific delivery to the TGN is required, it would be important to carry out the same experiments with the specific PKA inhibitors.

In a previous study (Calebiro et. al Molecular Endocrinology 2006 20:3196-211) we had already performed a detailed characterization of the role of PKA I/II isoforms on CREB phosphorylation and gene transcription in a well differentiated thyroid cell line (FRTL5), including the effects of selective PKA activators and inhibitors, Ht31 and silencing of RII α vs. RII β . This previous study, which is amply mentioned in the manuscript (see lines 210-212, 216-219, 220-222) clearly demonstrated that the presence and correct localization of PKA-II β (via AKAPs) is required for the effects of TSH on CREB phosphorylation and gene transcription.

Moreover, based on the reviewer's suggestion, we have now evaluated the effects of PKA I/II inhibitors on the induction of early genes in primary thyroid cells. Our new results (see new **Fig. 4f** and lines 216-220) show that the PKA II inhibitor also partially impairs the induction of early genes. These results are consistent with the effect we observe on CREB phosphorylation (see **Fig. 4e**), although the effect on CREB phosphorylation is somehow stronger, possibly due to signal amplification (i.e. the observed residual activation of CREB might still be sufficient to cause a partial induction of early genes).

We would also like to point out that, although experiments with these pharmacological inhibitors were not requested by any of the reviewers, we thought that repeating the key experiments of our Molecular Endocrinology study in primary thyroid cells would have been a nice addition to our previous revision. However, we feel that performing again all experiments in primary thyroid cells would be a repetition of our previous study and we don't believe that this would add much to our main conclusion that retrograde trafficking and local cAMP/PKA signaling in the TGN/Golgi are required for efficient TSH signaling.

We would also like to mention that, in general, we have given priority to CREB phosphorylation (over gene transcription) as we believe that looking at phosphorylation of CREB, which is directly phosphorylated by PKA, is more sensitive (see possible issue with amplification) and less prone to potential indirect effects of pharmacological treatments (see also discussion, lines 370-374) than evaluating downstream gene transcription.

3. The argument that delivery to the retromer compartment is essential could be further strengthened by demonstrating that Nb37 does not co-localise with EEA1 or indeed rab5.

We thank the reviewer for this important suggestion. Based on this, we have performed additional experiments in which we cotransfected primary mouse thyroid cells with Rab5-GFP and Nb37-SNAP, followed by stimulation with fluorescent TSH. Consistent with the apparent lack of triple colocalization between internalized TSH, Rab5 and Gs protein (see **Fig. 2b**), we found virtually no active Gs protein on the Rab5 positive early endosomes containing internalized TSH. Representative images and a detailed quantification (both in **new Fig. 3f**) have been added to the newly revised manuscript (see also lines 181-185).

Minor points: The title of the manuscript refers to trafficking to the TGN whereas in the rebuttal the authors state: 'We agree with the reviewer that we cannot rule out that G proteins get at least partially activated in the RCC before reaching the TGN, but we think this doesn't change at all the conclusion of our manuscript and is actually only a rather semantic issue as the separation between the retromer-coated domains on the endosome and the TGN is only virtual.'

Actually, although apparently semantic, it is important to define as accurately as possible the nature of intracellular compartments that are involved in signalling. The reality is that the environment of the TGN is likely to be quite different from that of retromer coated domains on endosomes and this is important to bear in mind for any future mechanistic studies. Hence the title should reflect this distinction.

We thank the reviewer for these comments. What we meant with our previous title is that Gs-protein signaling at intracellular sites begins once the internalized TSH receptor enters the RCC (as also depicted in the scheme of **Fig. 8**), which rapidly brings it to the TGN. As mentioned in the discussion (see e.g. lines 361-366) and pointed out by the reviewer, we agree that we cannot distinguish the effects of Gs protein signaling at the RCC before and after reaching the TGN. However, as also mentioned in the discussion (see lines 311-314), we think that the TGN (with the nearby Golgi) is the most likely critical site of

signaling by internalized TSH receptors, as it is the only place where we simultaneously observe the presence of internalized TSH/TSH receptor, active Gs-protein and PKA II.

Based on the reviewer's suggestion and these considerations, we have modified the title to "Internalized TSH receptors en route to the TGN induce local Gs-protein signaling and gene transcription", to more clearly state that Gs-protein signaling occurs once the receptors are committed to the TGN but might still be on their way to the TGN (i.e. they are in the RCC). We think that this title is correct, while being concise as required by *Nature Communications* and sufficiently clear for non-specialists.

2. On a similar theme, I suggest that the authors might consider calling the second and third phases of PKA activation 'later' rather than 'delayed' which suggests a defect whereas in fact they are arguing that these later waves are the physiologically relevant ones.

We thank the reviewer for underlying this potential source of confusion. Based on the reviewer's suggestion, we have replaced "delayed" with "late" phase or response. Still, in the description of the fittings, we have left the term "delay", as this describes correctly the observed kinetics. Indeed, the second and third phases come with a delay (temporal gap) from the first one, which presumably reflects some "rate-limiting" steps along the internalization/trafficking/intracellular signaling path.

Reviewers' Comments:

Reviewer #3 (Remarks to the Author):

The authors have done a good job addressing the issues raised.